# Natural Sources of Food Colorants as Potential Substitutes for Artificial Additives

**DOI:** 10.3390/foods12224102

**Published:** 2023-11-12

**Authors:** Erika N. Vega, María Ciudad-Mulero, Virginia Fernández-Ruiz, Lillian Barros, Patricia Morales

**Affiliations:** 1Departamento de Nutrición y Ciencia de los Alimentos, Facultad de Farmacia, Universidad Complutense de Madrid, Plaza Ramón y Cajal, s/n, 28040 Madrid, Spain; erinino@ucm.es (E.N.V.); mariaciudad@ucm.es (M.C.-M.); vfernand@ucm.es (V.F.-R.); 2Centro de Investigação de Montanha (CIMO), Instituto Politécnico de Bragança, Campus de Santa Apolónia, 5300-253 Bragança, Portugal; lillian@ipb.pt; 3Laboratório Associado para a Sustentabilidade e Tecnologia em Regiões de Montanha (SusTEC), Instituto Politécnico de Bragança, Campus de Santa Apolónia, 5300-253 Bragança, Portugal

**Keywords:** natural colorants, azo colorants, anthocyanin, betalain, carotenoid, chlorophyll, natural sources

## Abstract

In recent years, the demand of healthier food products and products made with natural ingredients has increased overwhelmingly, led by the awareness of human beings of the influence of food on their health, as well as by the evidence of side effects generated by different ingredients such as some additives. This is the case for several artificial colorants, especially azo colorants, which have been related to the development of allergic reactions, attention deficit and hyperactivity disorder. All the above has focused the attention of researchers on obtaining colorants from natural sources that do not present a risk for consumption and, on the contrary, show biological activity. The most representative compounds that present colorant capacity found in nature are anthocyanins, anthraquinones, betalains, carotenoids and chlorophylls. Therefore, the present review summarizes research published in the last 15 years (2008–2023) in different databases (PubMed, Scopus, Web of Science and ScienceDirect) encompassing various natural sources of these colorant compounds, referring to their obtention, identification, some of the efforts made for improvements in their stability and their incorporation in different food matrices. In this way, this review evidences the promising path of development of natural colorants for the replacement of their artificial counterparts.

## 1. Introduction

Color is one of the most important organoleptic characteristics of food products since it is one of the parameters that gives the consumer an idea about the food’s acceptability, its composition, flavor and/or freshness [1]. According to the Food and Drug Administration (FDA), a food colorant is “any dye, pigment, or substance which when added or applied to a food, drug or cosmetic, or the human body, is capable (alone or through reactions with other substances) of imparting colour” [2].

Colorants are added to food products for different reasons such as restoring the original color lost due to the influence of exposure to light, temperature variations, moisture and/or storage conditions; enhancing the natural color or adding color to colorless products [2,3].

Color additives can be classified as subject to certification and exempt from certification. First are colorants that are obtained synthetically, while second comprise colorants that are obtained from vegetables, minerals, or animals; therefore, this classification is the same as artificial and natural colorants. Another way of categorizing them is as straight colors, lakes, and mixtures. Straight colors refer to colorants that have not been mixed or chemically reacted with any other substance (e.g., FD&C Blue No. 1). Lakes for food use refers to straight colors that have been chemically reacted with precipitants and substrates such as aluminum cation as the precipitant and aluminum hydroxide as the substratum (e.g., Blue 1 Lake). Finally, mixtures are colorants generated by mixing a color additive with one or more color additives or non-colored diluents without chemical reaction [2,4].

Currently, artificial food colorants are widely use by the food industry, among others, in children’s products, since they have high intensity, stability, uniformity of color, and are cheap. However, their use has been controversial since 1976 when Feingold [5] reported a quick improvement in the behavior of children with attention deficit and hyperactivity disorder (ADHD) due to the elimination of the intake of artificial food colorants (mostly azo colorants) and flavorings. Since then, several studies have been published, the most well-known being the one provided by McCann [6], who evidenced the increase in the incidence of ADHD including inattention, impulsivity, and overactivity not only in children with extreme hyperactivity, but also in the general child population due to the consumption of artificial food colors (AFCs) and other additives such as sodium benzoate preservative (E 211). It is important to mention that a few studies regarding the effect of artificial food colorants in adults have been completed, such as that of Murdoch et al. [7], who evidenced a significant increase in the histamine levels of adults after the consumption of 150 mg of tartrazine; Di Lorenzo et al. [8], who evidenced symptoms of allergy in patients with chronic urticaria to different additives including tartrazine and erythrosine, in response to mixture of additives; Pestana et al. [9], who did not find evidence of statistical differences between the reaction to tartrazine and the placebo in 26 adults with allergic pathologies such as rhinitis, asthma or urticaria; and Park et al. [10], who tested a mix of seven food additives that included amaranth, erythrosine, tartrazine and sunset yellow FCF in 54 adults with allergic diseases evidencing that there were no significant differences in the reactions to the mix of additives and the placebo. However, the vast majority of studies on the effect of artificial food colorants have been carried out on children as they are more affected than the rest of the population, mainly due to the lack of control in their diet and to the fact that they are highly attracted by colors and present a higher consumption of processed sweet products such as candies or ice creams [11].

Consequently, the responsible entities, EFSA (European Food Safety Agency) in the European Union and the FDA (Food and Drug Administration) in the United States of America, have carried out different reviews of the available studies in order to evaluate the safety of azo colorants, among other additives. On one hand, the most important communications have been the evaluation carried out in 2009 in the European Union by the EFSA where only a decrease in the acceptable daily intakes (ADI; the amount of the substances that a person can consume daily without risk) of three of these problematic colorants (quinoline yellow, E104; sunset yellow FCF, E110; and Ponceau 4R, E124) was established [12,13,14]; and the statement release in 2013 by EFSA, where it was assured that a revaluation from the ADI of any of the azo colorants was not necessary and it was recommended that new tests should be carried out related to genotoxicity [15]. On the other hand, in 2011, the Food Advisory Committee from the FDA from the United States concluded that a relationship between children’s consumption of AFCs and behavioral effects had not been established [16].

However, the scientific community has continued working to establish the possible harmful effects of AFCs, having evidenced so far the potential of azo colorant to alter the normal functioning of the kidneys and liver, generation of reactive oxygen species, induction of hypersensitivity [11], proinflammatory responses [17] and, principally, an effect on ADHD [18]. In addition, in recent years, the use of AFCs in the food industry has exponentially increased, to the point that in some categories, mainly in children’s products such as cakes, chips, chocolates, ice creams, drinks, etc., all products contain at least one AFC, and it should be noted that there have even been cases where the product exceeded the allowed level [18,19,20]. On the whole, the different results have evidenced that is highly possible that many children may be consuming higher amounts of AFCs than previously thought [21].

As a result of all the evidence regarding the problems that AFCs can generate, as well as the growing interest of the general population in healthier and more natural food, it has been necessary to provide a revaluation of the AFCs ADI, or the approval of their use, and at the same time, to acknowledge the development of natural colorants that can be used instead of AFCs obtaining the same or even better results in the final product.

For the present review, scientific articles, scientific opinions and regulations from official databases such as PubMed, Scopus, Web of Sciences and ScienceDirect were consulted and evaluated with specific reference to sources of natural colorants, evaluation of colorant capacity, extraction of pigments from natural sources and extraction of each main colorant compound, anthocyanins, betalains, carotenoids and chlorophylls. Scientific studies published in the last 15 years (2008–2023) were taken into account in this review. Therefore, this review summarizes and compiles various food sources, their composition, extraction methods and stabilization methodologies, as well as the inclusion of several colorants obtained from these natural sources in different food matrices, highlighting the importance of the development of better and more stable natural colorants, as well as some of the drawbacks and shortcomings of the current colorants developed to date.

## 2. Principal Colorant Compounds Found in Natural Matrices

Among the compounds present in plants that show coloring properties, there are four main groups, anthocyanins, which have the widest range of colors presenting hues from blue to red; betalains, for which the color could be red-violet, yellow-orange color, and mainly pink; carotenoids, which can range from red to yellow; and chlorophylls, which are mostly green but also can present blue tones. All these compounds, in addition to their colorant capacity, exhibit different bioactive properties evidencing health benefits with their consumption [22].

### 2.1. Anthocyanins

Anthocyanins are pigments, for which the color can vary from blue to red. They derive from flavonoids and their base structure is an anthocyanidin which is constituted by two hexanes linked by three carbon atoms that form an oxygenated heterocycle; this structure presents a positive charge and is also known as a flavylium ion. When this anthocyanidin is in its glycosylated form, linked to a sugar, it is known as an anthocyanin (Figure 1) and this sugar can be bonded in different positions, mainly position C3 or C5 of the first ring. In addition, anthocyanin can be hydroxylated and/or methoxylated in different positions. This chemical structure of conjugated double bonds provides them with strong colors as well as high antioxidant properties through the scavenging of free radicals [23,24,25].

More than 700 different anthocyanins have been reported [26,27], which are differentiated from each other via the number and position of the hydroxy and methoxy substitutions, and the type and position of the sugar linked to the anthocyanidin [24,28]. These variations provide them with different colors, stability, bioavailability, and health effects [24,29].

Twenty-seven anthocyanidins have been observed in nature; however, cyanidin, delphinidin, pelargonidin, peonidin, malvidin, and petunidin represent 90% (50%, 12%, 12%, 12%, 7%, and 7%, respectively) of the anthocyanidins found in plants, being cyanidine-3-glucoside which is the predominant anthocyanin. The main edible sources of anthocyanins are berry fruits, including blueberry (*Vaccinium corymbosum* L.), bilberry (*Vaccinium myrtillus* L.), and strawberry (*Fragaria vesca* L.), among others. In addition, they can also be found in other foods, such as red sweet potato (*Ipomoea batatas* L.) and purple corn (*Zea mays* L.) [28]. Furthermore, different possible health benefits have been related to anthocyanins, such as possessing anti-inflammatory, anticancerogenic, antimutagenic and cardioprotective effects, being regulators of total cholesterol, LDL, HDL, and triglyceride levels, anticoagulants, and providing help in the prevention of neurological and cognitive alterations [24].

Anthocyanins are one of the biggest types of natural pigment; hence, they have been highly studied for the generation of natural colorants, as an alternative to artificial one. Nevertheless, one of the biggest problems to overcome is the low stability that anthocyanins present, since they can be affected by light, temperature, pH, metal ions, enzymes, oxygen, and co-pigments [30].

### 2.2. Betalains

Betalains are water soluble nitrogen-containing pigments with a core protonated structure commonly known as betalamic acid [4-(2-oxoethylidene)-1,2,3,4-tetrahydropyridine-2,6-dicarboxylic acid that can be found as betacyanins or betaxanthins, which have a pink–violet and yellow–orange color, respectively. Up to now, 75 different betalains have been identified, wherein, 51 of them are betacyanins. Therefore, betacyanins are the most commonly found. These pigments belong to 17 families of the order *Caryophyllales*. They are present in fruits, flowers, leaves, and roots, with betanin being the most common structure. On the other hand, betaxanthins can also be found in tubers, but not as much in leaves, with proline-betaxanthin being the most common [31,32]. The main edible sources of betalains are red beet roots (*Beta vulgaris* L.), amaranth (*Amaranthus* sp.) cacti (*Opuntia* sp.) fruits, and dragon fruit (*Hylocereus* sp.), among others. Different bioactivities have been related to betalains such as antioxidant, antibacterial, antifungal, antiprotozoal, and anticancer properties. Some of these effects, particularly the antioxidant properties, are related to the chemical structure of betalains, as they have a phenolic group and a cyclic amine, which are excellent electron donors and exhibit antiradical activities [32].

Betalains are also characterized by their reduced stability, as these pigments are thermolabile and affected by light, high water activity, pH under 3 or above 7 and oxygen. For these reasons, recent research has focused on the preservation of the chemical structure and the functionality of betalains, by using different techniques such as complex formation, copigmentation and encapsulation [33].

### 2.3. Carotenoids

Carotenoids are lipophilic pigments that go from yellow to red. They are composed of a C_40_ skeleton which contains polyene groups with end groups at both ends of the chain (Figure 1). Carotenoids can be classified in two categories according to their composition, namely carotenes and xanthophylls. Carotenes are hydrocarbons and include α-carotene, ß-carotene or lycopene. Xanthophylls contain functional groups with oxygen and include lutein, zeaxanthin or astaxanthin. These pigments can be found in different natural sources such as fruits, seed, roots and flowers, fulfilling functions as antioxidants, color attractants or hormone plant precursors. About 50 different carotenoids have been described in common human foods, including tomatoes (*Solanum lycopersicum* L.), persimmon (*Diospyros kaki* L.) and Gac fruit (*Momordica cochinchinensis* Spreng.), these being some of the most representative sources. Regarding their beneficial effect on human health, some beneficial properties have been related in relation to ocular, cardiovascular and fatty liver diseases, liver fibrosis and cancer. The majority of these potential health benefits have been attributed to the anti-inflammatory and immunomodulatory activity, and the regulation of cell cycle/apoptosis exhibited by these molecules [34,35].

Regarding the stability of these pigments, carotenoids are susceptible to various degradation and isomerization reactions, as was mentioned, above which cause their discoloration and can lead to a decrease and thus reduction in their biological activity. In particular, the stability of carotenoids is mainly affected by thermal processes, as well as by the presence of oxygen or light [36].

### 2.4. Chlorophylls

Chlorophylls are green natural colorants present in a wide number of green fruits and vegetables. Spinach, alfalfa, grass and nettles are natural sources of chlorophylls. These pigments are constituted of a porphyrin ring with a magnesium atom as the central metal (chlorophyll a and b) or without a central metal (e.g., phaeophytin a and b, pheophorbide a and pyropheophytin a), with an isocyclic five-membered ring and a chain of propionic acid esterified with phytol (Figure 1). To date, there are more than 100 chlorophylls or chlorophyll derivatives reported, with chlorophylls a and b being the most common in plants, which are found in a 3:1 proportion, respectively [37,38,39]. Chlorophylls present a significant antioxidant activity through chelation of reactive ions and scavenging of free radicals preventing DNA damage and lipid peroxidation, as well as antimutagenic and antigenotoxic activity through the prevention of mutagen migration and its coupling to DNA by the formation of a chlorophyll–mutagen complex which facilitates the degradation of the mutagen [37,40].

Chlorophylls are highly sensitive pigments that could be degraded by several factors, such as oxygen, light, heat, acids, and enzymes, causing a color change from green to brown, mostly due to the substitution of the central magnesium ion by two hydrogens, losing the green color characteristic of the presence of the magnesium ion [37,41].

## 3. Source of Natural Colorants as a Potential Replacers of Artificial Food Colorants

Currently, the European Food Safety Authority (EFSA) and the Food and Drugs Administration (FDA) have approved the use of different extracts from plants as colorants, as can be seen in Table 1, with the principal colorant compounds, anthocyanins, carotenoids, and chlorophylls being present, so there is a wide variety of colors. However, some of them are limited in their use to exact types of products, e.g., butter pea flower extract, which in the United States can only be used in beverages, juices, ice cream and candies, or copper complex of chlorophylls, which can only be used in citrus-based dry beverage mixes. Moreover, there is a bigger problem with these natural colorants than the restrictions of use to specific products, their low stability and therefore the small range of use, as these natural colorants can be affected by pH, temperature and light, among others, that make their use difficult in variety of products.

### 3.1. Source of Natural Red-Purple Colorants as a Potential Replacers of Artificial Colorants

Among the artificial red colorants, azorubine or carmoisine (E122), Ponceau 4R (E124) and FD&C Red No. 40 or Allura red AC (E129), are the most used by the food industry, being all of them azo colorants [44]. In 2009, the use of different colorants, including azorubine, Ponceau 4R and Allura red AC, was reassessed by the EFSA committee due to the evidence of possible consumption above the acceptable daily intake (ADI: 4, 0.7 and 0.7 mg/kg body weight per day, respectively) at the maximum level of use. In the case of azorubine and Allura red AC, consumption above the ADI was evidenced in children from 1 to 10 years old. As a result, several maximum permitted levels were decreased [45,46].

In the same way, with Ponceau 4R, the consumption above the ADI was evidenced in either 1 to 10 years old children or adults. Therefore, a refined exposure assessment was carried out and many maximum permitted levels were withdrawn or decreased [47]. However, neither the use of azorubine nor Ponceau 4R as a food colorant is approved by the FDA.

These food additives are food colors with a combined maximum limit according to the European Regulation (EU) nº 1129/2011 and therefore the use conditions of these food additives in different food categories and the maximum level permitted were established in this regulation [48].

Different vegetables, fruits and cereals can be considered as a potential source of natural red colorants (Table 2), which could replace the red artificial colorants. To attain this purpose, anthocyanins play an important role, as they present different hues of red, being the most studied pigments for the development of natural red food colorants. As food additives, Anthocyanin (E-163) is a natural food colorant approved in the European Union which is mainly obtained from purple carrots, grapes, radish and red cabbage, among others. It is a food color authorized at quantum satis according to the European Regulation (EU) nº 1129/2011 [48].

Regarding other food sources of this colorant, banana (*Musa X paradisiaca* L.) bracts, which are an abundant by-product of banana production, could be a good source of anthocyanin; its extraction was carried out by maceration with acidified methanol. Seven different anthocyanins were identified in the extract, with cyanidin-3-rutinoside being the main anthocyanin (80%) [49]. It was also possible to obtain 32 mg of anthocyanins per 100 g of bracts, which the authors determined as a significant amount. Later, Lestario et al. [50] evaluated the effect of 1% of tartaric acid, 1% of citric acid and 1% of acetic acid as solvents in the maceration extraction, evidencing that 1% of tartaric acid was the best option, which allowed 2.45 mg of anthocyanins to be obtained per 100 g of bracts. Most recently, the bracts of the banana ABB (*Musa acuminata x balbisiana*, ABB group) species have been studied, which is the most commercial type of banana grown in different countries, especially in Asia.

Different conditions for ultrasound-assisted extraction such as extraction solvent (40, 50 and 60% ethanol), sonication temperature (40, 50 and 60 °C) and s/L ratio (5, 10 and 15) were evaluated through response surface method (RSM) by Begum and Deka [51], finding that a relation solvent: solute of 15:0.5, 53.97% ethanol as solvent and 49.39 °C were the best conditions generating a total anthocyanin amount of 56.98 mg/100 g. Then, the anthocyanin extract was microencapsulated by spray-drying with maltodextrin as the carrier with 96.15% of encapsulation efficiency. Moreover, the stability of the encapsulated pigment was evaluated for 21 days, through the anthocyanin content, which varied from 57.64 mg/100 g to 32.30 mg/100 g, showing the highest change at 14 days of storage. Also, the color parameters expressed in the CIELAB model which denominate color with three parameters, *L**, *a** and *b**, where *L** indicates the lightness, *a** is the red (+)/green (−) coordinate and *b** is the yellow (+)/ blue (−) coordinate, were estimated showing values of *L**: 61.38, *a**: 21.53, *b**: −0.08, where *L** and *a** parameters were stable until 14 days and just showed significant changes at 21 days of storage. However, *b** parameter change significantly in the first 7 days of storage generating a change in the chroma and hue.

In order to enhance the stability of the extract, the same authors extracted by ultrasound the dietary fiber from banana bract and generated a dietary fiber–anthocyanin formulation in different proportions (control, 2:1, 3:1 and 4:1). The best anthocyanin content was achieved with 3:1 proportion (41.64 mg/100 g), as well as the lightest color (*L**: 50.42, *a**: 9.12, *b**: 8.14). In addition, the formulated colorants were stored under accelerated conditions at 35 °C and 75% RH where a fast degradation was observed during the first four weeks followed by a gradual degradation. Additionally, the degradation was lower in the sample with 3:1 proportion [52].

Red onion (*Allium cepa* L.) is another vegetable that has been studied in-depth lately due to the high amount of solid waste that it generates. Mourtzinos et al. [53] determined through LC-MS that cyanidin-3-*O*-glucoside was the predominant anthocyanin. In addition, they evaluated a different mix of solvents (40–100% of water, 0–60% of glycerol and 0–13% of cyclodextrin), as well as different temperatures (40–80 °C) for the extraction of anthocyanins by magnetic stirring at 600 rpm for 240 min. Taking into account the total pigment yield, it was concluded that the best conditions were a solvent mix with 60% of glycerol and 13% of cyclodextrin at 80 °C, since it generated a total pigment yield of 3.13 mg cya-3-gluE/g (mg cyanidin-3-*O*-glucoside equivalent per g, dw). Later, Krithika et al. [54] determined that the best conditions for extraction of anthocyanins through microwave were 5 min, 700 W of power, 75% ethanol as extraction solvent and a solvent feed ratio of 20 g/mL. Moreover, the anthocyanin extract obtained (21.99 mg of monomeric anthocyanin/L) was stabilized by three different ways where the use of buffer combination (sodium carbonate and sodium bicarbonate, pH: 1) generated the most reddish and stable samples, co-pigmentation with gum Arabic-generated stable but colorless samples, and dark brown anthocyanins were obtained by pan coating.

Red calyces of *Hibiscus sabdariffa* L. are another great source of anthocyanins with the majority being delphinidin-3-*O*-sambubioside [55]. Through response surface methodology, Pinela et al. [56], determined that the best method for obtaining an extract rich in anthocyanins from red calyces of *Hibiscus sabdariffa* was through ultrasound under conditions of 296.6 W of power and 39.1% ethanol/water as solvent for 26.1 min. These results were achieved after comparing heat and ultrasound-assisted extraction methods with different combinations of time, ultrasonic power, and ethanol proportion. Applying the determined conditions, it was possible to recover 223.83 mg of anthocyanins per gram of dried plant material and 51.76 mg in 1 g of extract. Later, Escobar-Ortiz et al. [57] analyzed the best conditions for increase the extraction yield from *H. sabdariffa*, concluding that the temperature, time, type or percentage of acid and the water: ethanol ratio did not affect the yield of the anthocyanin extraction; however, with the highest s/L ratio (1:8), this increased the extraction yield, obtaining 1100–1700 mg of cyanidine-3-sambubioside equivalent/100 g of sample. In addition, the stability of the different extracts was evaluated at different temperatures for 45 days showing that the extracts with the highest percentage of ethanol (80%) avoided the degradation of anthocyanin in 50%. In addition, a relation between the stability of the anthocyanins and the presence of quercetin, myricetin, kaempherol-3-*O*-glucose, ellagic acid, and rutin was reported by the same authors. On the other hand, several studies have been completed in order to improve its stability over time through encapsulation; among the most recent studies, a high stability of an anthocyanin extract was achieved under low humidity conditions by microencapsulation with yeast hulls, which was maintained after ten weeks of storage. Moreover, any significant change in color parameters were evidenced under different conditions of temperature (5 or 37 °C) comparing the initial time (*L**: 19.79, *a**: 5,45, *b**: 2.56) and after ten weeks of storage (*L**: 19.68, *a**: 5,45, *b**: 2.78) [58].

Red cabbage (*Brassica oleracea* L.) has been also widely investigated due to the different colors that can present in a wide range of pH levels [59]. The best conditions for the anthocyanin extraction from this matrix have been studied by several authors, such as Chandrasekhar et al. [60], who carried out several extractions with different solvents (water, acidified water, ethanol, methanol, acetone and different mixes of them), obtaining a total anthocyanin content of 390.6 mg/L with a mixture of 50% (*v*/*v*) of ethanol and acidified water. Moreover, they obtained an anthocyanin extract free of sugar after the purification through non-ionic acrylic ester adsorbent (Amberlite XAD-7HP). Later, Ravanfar et al. [61] obtained a higher yield by ultrasound extraction under conditions of 100 W of potence, a pulse mode of 300 s ON: 30 s OFF and 15 °C for 90 min than by traditional percolation. On the other hand, there are several studies about the stabilization of anthocyanin extracts from red cabbage, with spray drying as the most studied methodology, evidencing high stability at temperatures between 40 and 90 °C with maltodextrin and gum Arabic mix (35: 15%) [62] and 140 and 160 °C with gum Arabic and polydextrose [63].

In purple or black carrot (*Daucus carota* L. sp. *Sativus var. atrorubens*), five different anthocyanins were identified through HPLC, the majority of which was cyanidin-3-xyloxyl-glucosil-galactoside acylated with ferulic acid [64]. Different authors have evaluated the best conditions for the stabilization of a red extract obtained from purple carrots, such as Ersus et al. [65], who evaluated different types of maltodextrins evidencing that the highest total anthocyanin content (630.92 mg cyanidin-3-galactoside/100 g) was obtained with Glucodry 210. Later, Assous et al. [64] evaluated different carriers for the stabilization of anthocyanin extracts through spray drying, evidencing that dextrin was better than soluble starch, cellulose and glucose. In addition, the influence of pH from 1.0 to 10.0 and different temperatures, from 40 to 100 °C for 30 min, have been evaluated in an encapsulated extract with dextrin, evidencing that the colorant was stable between pH 1.0 and 4.0 and at a temperature of 40–80 °C.

Regarding fruits, berries are one of the most studied matrices for obtaining natural colorants, mostly due to their intense color from red to purple and, therefore, their high content in anthocyanins. In addition, there is a huge variety of berries with amounts of anthocyanins that can vary from a few mg/100 g to almost 2000 mg/100 g of sample analyzed [66]. Buran et al. [67] extracted anthocyanins from blueberries by applying ultrasound to the mix of the fresh fruits with acidified hot water (s/L: 0.4 g/mL; 90 °C) followed by sonication (100% amplitude for 5 min). In addition, the adsorption capacity and desorption ratio of different resins (XAD4, XAD7HP, XAD761, XAD1180, FPX66) were tested in order to determine the best material for the concentration of the colorant, eliminating the sugars from the extract. FPX66 took the least time in reaching adsorption equilibrium and showed the highest adsorption efficiency, through this medium a yield of 0.80 g of concentrated extract was obtained from 100 g of fresh fruit, free of sugar. On the other hand, recent studies have been directed to the stabilization of the anthocyanins obtained from blueberries. An increase in stability of 76.11% after 30 days of storage was evidenced after the microencapsulation with carboxymethyl starch and xanthan gum (30/1), with a minimum change in the color, with *L**: 43.94, *a**: 0.68 and *b**: 1.76 being the extract color parameter and *L**: 45.87, *a**: 4.95 and *b**: 0.26 the encapsulated extract color parameters [68]. More recently, a high encapsulation efficiency and great physical properties of the microcapsules were evidenced with the use of whey protein isolate and casein as wall materials. Therefore, through in vitro digestion, these encapsulating agents allowed the inhibition of the rapid release and degradations of the anthocyanins [69]. Andean blueberry (*Vaccinium meridionale* Swartz) presents a total anthocyanin content of 747.6 mg/100 g, fw, where cyanidin-3-galactoside represents the majority [70]. For this fruit, 30 MPa, 313 K and 30 min were determined as the optimal condition for anthocyanin extraction through supercritical CO_2_-assisted extraction, generating 92% of material recovery [71]. Later, the suitability of maltodextrin as the encapsulant agent for an extract rich in anthocyanins from this fruit was evaluated, finding that using between 30 and 50% of this carrier of higher monomeric anthocyanin and polyphenol recoveries presented color parameters of *L**: 54.2 and 52.7, *a**: 36.6 and 38.0, *b**: 1.4 and 2.9, respectively [72].

In the same way, blackberry (*Rubus* spp.) has been widely studied for obtaining its extracts. Yamashita et al. [73] produced an extract rich in anthocyanins from blackberry pulp by-product using maceration at room temperature for 8 h. Then, the extract was encapsulated with 10 and 20 DE (dextrose-equivalent). A better anthocyanin retention (265.73 mg cya-3-glu/100 g, fw; 76.55%) was evidenced, acidity and hygroscopicity with 10 DE as the amount of microencapsulating matrix, as well as a darker color (*L**: 34.34; a*: 20.96; *b**: 6.51). In the same year, Weber [74] increased the shelf-life stability and maintained the color of a spray-dried blackberry extract (151 half-life days) through the addition of rutin (193 half-life days) and ferulic acid (158 half-life days), which are believed to create a complex combination and therefore generate a copigmentation.

The genus *Rubus* spp. contains several species, with *Rubus fruticosus* L. being one of the most representative and, therefore, more studied, which presents four anthocyanins, all of them derivates of cyanidin [75]. The stability of an extract rich in anthocyanins from *R. fruticosus* was increased by microencapsulation with maltodextrin through spray drying, which after seven days presented a total monomeric anthocyanin content of 126 mg of cyanidin-3-glucoside per 100 g at pH 2.0, which was 2.4 higher compared to the control (53 mg of cyanidin-3-glucoside per 100 g), and 33 mg of cyanidin-3-glucoside per 100 g at pH 5.0, a parameter that in the case of the control was not detected [76]. Fruits of *R. fruticosus* have shown one of the highest antioxidant activities among different wild fruits, (TBARS methodology: EC_50_ 25.3 µg/mL) [77]. Later, an increase was evidenced in the antioxidant activity through the OxHLIA (IC_50_ 250 µg/mL) methodology with the same microencapsulation conditions, as well as an increment in the antibacterial and antifungal activity [75].

*Morus nigra* L. is another type of blackberry with a high content of anthocyanins, principally cyanidin-3-*O*-glucoside and its derivates [75,78]. In the latest works, the influence of temperature and ultrasound amplitude were found as the most influential variables in the anthocyanin ultrasound-assisted extraction, determining 48 °C and 76% as the optimal conditions, respectively [79]. Moreover, a high stability of the color parameters and anthocyanin content was evidenced by encapsulation with maltodextrin and gum Arabic [75].

Blackcurrant (*Ribes nigrum* L.) contains 900 mg of total anthocyanin every 100 g of extract with four anthocyanins, two derivates of cyanidin and two derivates of delphinidin. It was evidenced that improvement in the extract stability through copigmentation with different phenolic acids was linked to the pH. Chlorogenic acid presented a higher stability at pH 3 allowing an anthocyanin retention of 86.3%; however, at pH 6, ferulic acid presented higher stability [80]. On the other hand, the optimal conditions for obtaining an extract rich in anthocyanins through ultrasound-assisted extraction (UAE) and enzyme-assisted extraction (EAE) were determined. A temperature of 5 °C, pH 4.97, ratio s/L 0.66 g/ L and 65% methanol in water as extraction solvent in the case of UAE, and 30 °C, pH 4.1, 10% ethanol in water as extraction solvent, 50% of amplitude and 91.0 units of enzyme per gram of sample in the case of EAE [81].

Haskap (*Lonicera caerulea* L.) is mainly found in northeastern Asia and Japan. Its purple fruits present a high quantity of anthocyanins, divided into six anthocyanins, with the majority being cyanidin-3-glucoside [82]. The ultrasound-assisted extraction of anthocyanins from *L. caerulea* was optimized through RSM determining 35 °C for 20 min, with the L/s rate of 4 mg/L and 80% ethanol with 0.5% of formic acid as the optimal conditions for obtaining 2273 mg cyanidin-3-glucoside equivalents/100 g, dw [83]. Later, the stability of an extract rich in anthocyanins from *L. caerulea* was improved through spray drying with maltodextrin and a mix of maltodextrin and gum Arabic; both encapsulants allowed the anthocyanin content (9790 mg/100 g of extract) and the color parameters to be maintained for 12 weeks with minimum changes [82].

Chokeberry fruits (*Aronia melanocarpa* (Michx.) Britton) or aronia present a high anthocyanin content, mostly cyanidin glucoside (98.4%) [84]. In order to optimize the extraction of the bioactive compounds, a better yield was evidenced with the use of acidified methanol in preference to other alcohols [85], as well as a higher anthocyanin yield (134 mg cyanidin-3-glucoside equivalent/100 g, fw) with the use of subcritical water extraction compared with pressed juice and hot water [84].

In 2011, Sivakumar et al. [86] studied the extraction of pomegranate (*Punica granatum* L.) rind through ultrasounds and magnetic stirring under different conditions, finding a 25% increase in anthocyanin yield when using ultrasound extraction under conditions of 80 W and 45 °C for 3 h, compared with magnetic stirring (control). The following year, Qu et al. [87] tested the storage conditions, in terms of pH and light presence, of an aqueous extract, rich in anthocyanins, of pomegranate marc from juice production; this extract was generated under conditions of 25 °C, s/L ratio of 50:1 (*w*/*w*) for 2 min and sterilized at 121 °C for 10 s. They concluded that the best storage conditions were in darkness and at a low pH (~3.5), under which, after 180 days, the sample preserved 67% of the total soluble phenolic concentration and 58% of antioxidant activity.

Dykes et al. [88] evaluated the color and the antioxidant capacity of different lines and hybrids of black sorghum (*Sorghum bicolor* L. Moench) due to its high content of 3-deoxyanthocyanidins, molecules that carry the black color. It was reported that, even though the lightest hybrids presented higher antioxidant activity due to the presence of condensed tannins, it was recommended to continue working in the darker line (Tx3362) due to the high content of 3-deoxyanthocyanidins; therefore, this presented the best dark red color (*L**: 34.2, *a**: 3.8, *b**: 2.8).

Another natural pigment, which could be used as a natural red colorant is lycopene. It is a carotenoid constituted of a 40-carbon-atom chain with 11 conjugated and 2 unconjugated double bonds. This compound is the main carotenoid found in tomato [89,90] and it has been reported that lycopene extracted from tomatoes or its by-product is comparable with the food coloring agent used in EU (E160d). With an acceptable daily intake (ADI) of 0.5 mg/kg bw/day, lycopene is a food color that has restricted uses (with a maximum level) according to European Regulation nº 1129/2011 [48,91]. Briones-Labarca et al. extracted lycopene from tomato pulp (*Solanum lycopersicum* L.) by using a high hydrostatic-pressure-assisted methodology. These authors observed that the pressures applied as well as the solvent mixture used have a significant influence on the lycopene extraction yield, proving that the optimal conditions were 450 MPa and 60% hexane as solvent mixture. Under these conditions, the concentration of lycopene in the obtained extract was 2.08 mg/100 g, fw [92]. Other authors determined the optimal conditions of microwave-assisted extraction for lycopene from tomato peels and concluded that this technique allowed around 13 mg of all-trans lycopene/100 g to be obtained, with ethyl acetate at 400 W for 1 min [93].

Among foods which could be used as sources of lycopene, guava has been investigated to obtain lycopene-rich extracts from its pulp and waste. The extraction process was performed by maceration assisted by ultrasound and the concentration of lycopene in the extracts was 135.0 mg/100 g (extract from pulp) and 76.64 mg/100 g (extract from waste). The high concentration of lycopene in the extract obtained from guava’s pulp resulted in the strongest red color, confirmed by high *a** values in CIELAB scale [94].

Lycopene extracts have also been obtained from watermelon through two techniques, obtaining concentrations from 10.92 (by recrystallization) to 48.1 μg/g (by thin-layer chromatography) [95]. Moreover, the use of lycopene extracts in the food and nutraceutical industries is being highly recommended due to their color and mostly for their health effects [96].

Carmine (E120) and carminic acid are also natural red food colorants, which are currently produced from scale insects’ cochineal (*Dactylopius coccus* Costa) [97,98]. These food additives are food colors with a combined maximum limit according to the European Regulation nº 1129/2011 [48]. In contrast to anthocyanins and lycopene (both natural food colorants extracted from vegetables), carmine and carminic acid are products of animal origin. Chemically, carminic acid is an anthraquinone, and carmine is the aluminum and calcium salt of carminic acid. As food colorants, these compounds are incorporated in several products, including ice-cream, yogurt, or fruit juice, among others. However, they could cause allergic effects in some people [99].

Regarding purple color, anthocyanins (E163) are also the main purple colorant used by food industry. These compounds are authorized as food additives in the European Union, and they have previously been evaluated by JECFA in 1982 and by the EU Scientific Committee for Food (SCF) in 1975 and 1997. The acceptable daily intake (ADI) established by JEFCA is 2.5 mg/kg bw for anthocyanins from grape skin, whereas SCF has not determined an ADI for them [100].

Anthocyanins are widely distributed in various fruits, and several fruit extracts are used as purple colorants. It is important to highlight that anthocyanins’ color is the result of the ratio of the three colored species in equilibrium at any specific pH. In this way, while at the lowest pH (<2), anthocyanins show red color; meanwhile, at a higher pH (6–7), they exhibited a purple color [101]. The extracts most often used for obtaining natural purple colorants are grape-skin extract (containing glucosides of peonidin, malvidin, delphinidin, and petunidin), and blackcurrant extract (containing cyanidin 3-rutinoside, delphinidin 3-rutinoside, cyanidin 3-glucoside, and delphinidin 3-glucoside). In the EU, it is not necessary to specify which food matrix (fruits or vegetables) has been used to obtain the food colorant E163. Thus, the composition of this food additive is not specified, as these compounds are usually present as anthocyanins glycosides [100].

There are several studies that investigate obtaining anthocyanin extracts from different fruits, with blueberry as one of the most explored. The results obtained by Zheng [102] showed that the optimal conditions for extraction of anthocyanins from blueberries through microwave-assisted extraction were 7 min, 47.1 °C, ethanol 55% as extraction solvent and a solid/liquid ratio of 29.5 g/L. These conditions generated a purple extract with an extraction rate of 73.73%. On the other hand, the optimal conditions for anthocyanin extraction from blueberry (*Vaccinium corymbosum* L.) through subcritical water extraction were also determined, obtaining 47 mg per 100 g, fw at 130 °C for 3 min and pH 1 [84].

The leaves of false shamrock (*Oxalis triangularis*) were chemically characterized by Pazmiño et al. [103] who reported a monomeric anthocyanin content of 2.42 g/100 g, where malvidin-3-rutinoside-5-glucoside was the majoritarian anthocyanin. Moreover, an attractive purplish color (*L**: 86.0, *a**: 7.9 and *b**: 1.4) was maintained at pH 3.5.

Another matrix that is highly studied is eggplant (*Solanum melongena* L.) skin due to its dark purple color and its high anthocyanin content. Its anthocyanin composition was evaluated in an aqueous acetone extract obtained by continuous stirring for 60 min, founding four derivates of delphinidin, with the majority being delphinidin-3-rutinoside. In addition, the stability of the extract was evaluated as rich in anthocyanins at different pH levels, evidencing a better stability under pH 2 (*L**: 65.83, *C**: 52.12, h: 8.92) (where C* is the chroma and determines the saturation and *h* is the hue and refers to the tone, both of which can be calculated as the mathematical relationship between *a** and *b**; *C**: (*a**^2^ + *b**^2^)^½^, *h*: arctan (*b**/*a**)) and a degradation in the color above pH 6 leading to a brownish hue (ΔE*(pH 1–pH 6): 50,38) [104]. Later, the best conditions related to s/L ratio, time, temperature, and ultrasound frequency for anthocyanin extraction, through ultrasound, with methanol and 2-propanol were evaluated by RSM. It was found that 54.4% methanol, 37 kHz, 55.1 °C and 44.85 min were the optimal conditions obtaining a total anthocyanin of 2410.71 mg cyanidin-3-glucoside/ kg [105]. The latest works have been directed to the stabilization of this colorant mostly by microencapsulation generated by spray drying where maltodextrin has been determined as one of the best carriers; in this case, maltodextrin maintained the extract color closest to the original (*L**: 25.59, *a**: 6.17, *b**: 2.48) [106]. However, stabilization by nanoencapsulation has also been studied with gelatin biopolymer with two concentrations (33.3 and 20 mg/mL) through an electrospinning process generating electrospun nanofibers with 92 and 94% of encapsulation efficiency, respectively [107].

In 2014, Tan et al. [108] generated a red-purple extract rich in anthocyanins from *Rhoeo spathacea* (Swartz) Stearn leaves. The extraction was carried out with 70% aqueous acetone + 1% formic acid and a s/L ratio of 10 mL of solvent per 1 g of leaves. Its stability was tested under 14 different pH levels (1.0–11.5), in the presence and absence of light and at different temperatures (8–25 °C) for six months. As a result, a great stability was evidenced in the absence of light, cold (8 °C), and between pH 1.0 and 6.0. This stability showed at a pH above 4 was attributed by the authors to the presence of anthocyanins with multiple acylated groups which helps the stability of these molecules.

Although fruits are the major sources of anthocyanins, some cereals also present a considerable content of these pigments. This is the case with black rice bran, where enzymatic extraction was used, then the dark purple extract obtained was mixed with 2% of maltodextrin and freeze dried. This colorant extract presented 37.10 mg/g of total phenolic content and 9.48 mg/g of total anthocyanin content. These values were nine and three times higher than the values of the black rice bran (3.86 and 3.31 mg/g, respectively). In addition, the color parameters were evaluated evidencing a dark red color (*L**: 39.85, *C**: 17.66, h: 12.60) [109].

**Table 2 foods-12-04102-t002:** Natural sources of red–purple colorants, their extraction conditions and color parameters.

Pigment	Natural Source	Pigment Content of Extract	Extraction Conditions	Color Parameters	Reference
Red color					
Anthocyanins	Banana (*Musa X paradisiaca* L.) bracts	32 mg of anthocyanins/100 g, being cyanidin-3-rutinoside the main anthocyanin (80%)	Maceration extraction: - solvent: 0.15% of HCl in methanol	Color parameters of extract: *L**: 86.8, *a**: 9.1 *b**: 8.9 *h*: 44.2, *C**: 12.7	[49]
		2.45 mg of anthocyanins/100 g	Maceration extraction: - solvent: 1% of tartaric acid in water - s/L ratio: 5 g in 30 mL	-	[50]
		56.98 mg of anthocyanins/100 g	Ultrasound-assisted extraction: - solvent: 53.97% ethanol in water - relation solvent/solute: 15/0.5 - temperature: 49.39 °C	Color parameters of encapsulated extract: *L**: 61.38, *a**: 21.53, *b**: −0.08, *h*: 0.19, *C**: 21.52	[51]
		41.64 mg of anthocyanins/100 g of dietary fiber–anthocyanin formulation	Ultrasound-assisted extraction of anthocyanin: - solvent: 53.97% ethanol in water - relation solvent/solute: 15/0.5 - temperature: 49.39 °C	Color parameters of dietary fiber–anthocyanin formulation: *L**: 50.42, *a**: 9.12, *b**: 8.14, *h*: 41.75, *C**: 12.22	[52]
	Red onion (*Allium atrorubens* S. Watson)	3.13 mg of cyanidin-3-O-glucoside equivalent/g, dw	Magnetic stirring at 600 rpm for 240 min: - solvent: 60% of glycerol and 13% of cyclodextrin in water - temperature: 80 °C - s/L ratio: 50 mL/g	-	[53]
		21.99 mg of monomeric anthocyanin/L	Microwave: - extraction solvent: 75% ethanol in water - solvent feed ratio: 20 g/mL - time: 5 min - power: 700 W	-	[54]
	Red calyces of roselle (*Hibiscus sabdariffa* L.)	51.76 mg of anthocyanins/g of extract	Ultrasound-assisted extraction: - extraction solvent: 39.1% ethanol/water - time: 26.1 min - power: 296.6 W	*-*	[56]
		1100–1700 mg of cyanidine-3-sambubioside equivalent/100 g	Solid–liquid extraction: - temperature: 35 °C and 75 °C - type of acid: acetic and citric - percentage of acid: 0.5 and 2.0% - time: 15 and 60 min - s/L ratio: 1/8 and 1/3 - water/ethanol ratio: 80/20 and 20/80	-	[57]
		1308 mg of anthocyanins/100 g	Anthocyanin extract microencapsulated with yeast hulls	Color parameter of encapsulated extract: *L**: 19.79, *a**: 5.45, *b**: 2.56, *h*: 0.97, *C**: 6.02	[58]
	Red cabbage (*Brassica oleracea* L. Var. Capitata f. Rubra)	390.6 mg of total anthocyanins/L	Extraction with 50% (*v*/*v*) of ethanol and acidified water: - s/L ratio: 1/2 - time: 1 min	Color parameters of extract: *L**: 25.04, *a**: 44.81, *b**: 13.01, *h*: 16.18, *C**: 46.67	[60]
		31.08 mg of anthocyanins/L	Ultrasound-assisted extraction: - ethanol/water: 1/1 - power: 100 W - pulse mode: 300 s ON: 30 s OFF - temperature: 15 °C - time: 90 min	-	[61]
	Purple or black carrot (*Daucus carota* L.)	630.92 mg cyanidin-3-galactoside/100 g	Maceration extraction: - solvent: ethanol: 1.5 N HCl (85:15 *v*/*v*) - time: 8 min	Color parameters of microencapsulated powders: *L**: 53.82, *a**: 29.16, *b**: 5.83, *h*: 11.31, *C**: 29.74.	[65]
		168.70 mg of anthocyanins/100 g, fw	Maceration extraction: - solvent: ethanol acidified with 0.01% citric acid. - s/L ratio: 20 g/100 mL - time: 5 min	-	[64]
	Blueberry (*Vaccinium corymbosum* L.)	175.9 mg/g extract	Ultrasound-assisted extraction: - solvent: hot water acidified (0.5% *v*/*v* acetic acid) (90 °C) followed by sonication: - amplitude: 100% -time: 5 min	-	[67]
		-	Anthocyanins with purity of 25% were purchased, then encapsulated with different combinations of carboxymethyl starch/xanthan gum	Color parameters of extract: *L**: 43.94, *a**: 0.68, *b**: 1.76, *h*: 68.84, *C**: 1.89	[68]
		747.6 mg of cyanidin-3-glucoside/100 g, fw.	Ultrasound-assisted extraction: - methanol 80% - sonicated for 20 min - room temperature	-	[70]
		371.5 mg cyanidin-3-glucoside/L	Crushed with juice press extractor	Color parameters of the extract encapsulated with maltodextrin (30 and 50%): *L**: 54.2 and 52.7, *a**: 36.6 and 38.0, *b**: 1.4 and 2.9, *h*: 2.2 and 4.3, *C**: 36.6 and 38.1, respectively.	[72]
	Blackberry (*Rubus* spp.: *Morus nigra* L.; *Rubus fruticosus* L.)	718.47 mg cyanidin 3-glucoside/100 g, fw	Maceration extraction: - s/L ratio: 1:3 (m/v) - temperature: room temperature - time 8 h - in absence of light	Color parameters of microencapsulated extract: *L**: 34.34; *a**: 20.96; *b**: 6.51, *h*: 0.3, *C**: 21.94	[73]
		- 247 mg of cyanidin-3-glucoside/100 g (at pH 2.0) - 161 mg of cyanidin-3-glucoside/100 g (at pH 5.0)	Mechanical stirring extraction: - solvent: ethyl alcohol 80% - time: 48 h - rotavoparated at 65 °C - ratio: 500 mg/mL	-	[76]
	Bilberry (*Vaccinium myrtillus* L.)	210.06 mg cyanidin-3-glucosido/ 100 g, fw	Freeze-dried and milled	Color parameters of the dry fruit: *L**: 25.66, *a**: 17.02, *b**: 7.01, *h*: 22.39, *C**: 18.41	[110]
	Blackcurrant (*Ribes Nigrum* L.)	900 mg of total anthocyanins/100 g of extract	Maceration extraction: - solvent: 0.1% HCl in methanol - time: 1 h	-	[80]
		Ultrasound-assisted extraction: 2199 mg anthocyanin/100 g Enzyme-assisted extraction: 2164 mg anthocyanin/100 g	Ultrasound-assisted extraction: - extraction solvent: 65% methanol in water - ratio s/L: 0.66 g/L - temperature: 5 °C - pH 4.97 Enzyme-assisted extraction: - extraction solvent: 10% ethanol in water - 50% of amplitude - 91.0 units of enzyme per gram - temperature: 30 °C - pH 4.1	-	[81]
	Haskap (*Lonicera Caerulea* L)	2273 mg cyanidin-3-glucoside equivalents/100 g, dw	Ultrasound-assisted extraction optimized through RSM: - temperature: 35 °C - time: 20 min, - L/s rate: 4 mg/L - extraction solvent: 80% ethanol with 0.5% of formic acid	-	[83]
	Chokeberry fruits (*Vaccinium Corymbosum* L.)	134 mg cyanidin-3-glucoside equivalent/100 g, fw	Subcritical water extraction: - solvent: water and 1% citric acid - temperature: 190 °C - time: 1 min	-	[84]
	Sweet cherry (*Prunus avium* L.)	4.67 mg cyanidin-3-gucloside/100 g, fw	The skin was freeze-dried and milled	Color parameters of the dry fruit: *L**: 62.88, *a**: 12.68, *b**: 24.11, *h*: 62.26, *C**: 27.24	[110]
	Strawberry (*Fragaria vesca* L.)	25.56 mg cyanidin-3-gucloside/100 g, fw	Freeze-dried and milled	Color parameters of the dry fruit: *L**: 56.66, *a**: 23.59, *b**: 13.33, *h*: 29.47, *C**: 27.09	[110]
	Hawthorn (*Crataegus monogyna* Jacq.)	251.7 mg cyanidin-3-gucloside/100 g, dw	The skin was freeze-dried and milled	Color parameters of the dry skin: *L**: 57.70, *a**: 16.15, *b**: 20.65	[111]
	Whitebeam (*Sorbus aria* (L.) Crantz	33.7 mg cyanidin-3-gucloside/100 g, dw	The skin was freeze-dried and milled	Color parameters of the dry skin: *L**: 71.02, *a**: 9.06, *b**: 29.34	[111]
	Black sorghum (*Sorghum* Moench)	-	Maceration extraction: - solvent: 1% of HCL in methanol - time: 2 h - s/L ratio: 0.1–0.5 g/25 mL	Color parameters of Sorghum kernels: *L**: 34.2, *a**: 3.8, *b**: 2.8.	[88]
Carotenoids: Lycopene	Tomato (*Solanum lycopersicum* L.) pulp	2.08 mg lycopene/100 g, fw	High hydrostatic pressure-assisted extraction: - pressure: 450 MPa - solvent: 60% hexane	-	[92]
	Tomato (*Solanum lycopersicum* L.) peel	13.592 mg all-trans-lycopene/100 g extract	Microwave-assisted extraction: - solvent: ethyl acetate - time: 1 min - microwave power: 400 W - energy: 24 kJ equivalent for 1 min	-	[93]
	Tomato (*Lycopersicon esculentum* Mill.)	66.019–118.98 mg/kg, fw	Accelerated solvent extraction: - adjuvants: NaCl and paraffin oil - dehydrating agent: diatomaceous earth powder	-	[92]
	Guava (*Psidium guajava* L. cv. ‘Pedro Sato’)	135.0 mg/100 g (extract from pulp); 76.64 mg/100 g (extract from waste)	Bath-type Ultrasound-assisted extraction: - solvent: ethyl acetate - frequency: 40 kHz - nominal power: 300 W - temperature: 25 °C - time: 30 min	-	[94]
	Watermelon (*Citrullus lanatus* (Thunb.) Matsum & Nakai)	1.092–4.81 μg/g	Maceration extraction: - solvent: methanol - time: 1 h - temperature: 30 °C	-	[95]
Purple color					
Anthocyanins	Blueberry (*Vaccinium corymbosum* L.)	47 mg/100 g, fw	Subcritical water extraction: - solvent: water and 1% citric acid - temperature: 130 °C - time: 3 min	-	[84]
	False shamrock leaves (*Oxalis Triangularis* S.-Hil)	195 mg anthocyanins/100 g.	Maceration extraction: solvent: 0.15% HCl in methanol.	Color parameters (at pH 3.5): *L**: 86.0, *a**: 7.9 and *b**: 1.4, *h*: 9.8, *C**: 8.0	[103]
	Eggplant (*Solanum melongena* L.) skin	45.01 mg of anthocyanins/100 g	Maceration extraction: - solvent: trifluoroacetic acid in water/acetone (30/70) - time: 60 min - ratio s/L: 1:2	Better stability of color at pH 1 (*L**: 65.83, *C**: 52.12, h°: 8.92).	[104]
		2410.71 mg cyanindin-3-glucoside/kg	Ultrasonic-assisted extraction: - solvent: 54.4% methanol - frequency: 37 kHz - temperature: 55.1 °C - time: 44.85 min	-	[105]
		-	Maceration extraction: - solvent: distilled water - temperature: 80 °C - time: 40 min - ratio s/L: 5 g/100 mL	Color parameters of encapsulated extract: *L**: 25.59, *a**: 6.17, *b**: 1.05, *h*: 9.66, *C**: 6.26	[106]
	Black rice (*Oryza sativa* L.) bran	948 mg of total anthocyanins/100 g	Enzymatic extraction. - 37 °C, pH: 7.5, protease, 30 min - 65 °C, pH: 6.9, α-amilasa, 60 min - 85 °C, 10 min	Color parameters: *L**: 39.85, *C**: 17.66, h: 12.60	[109]

*L**: lightness, *a**: red/green coordinate; *b**: yellow/blue coordinate, *h*: hue, *C** chroma, s/l ratio: solid/liquid ratio.

### 3.2. Source of Natural Pink Colorants as a Potential Replacers of Artificial Colorants

Erythrosine, E-127 or FD&C Red No. 3, a xanthene dye, is the principal pink artificial colorant used by industry. It is not an azo colorant; however, it has been evaluated twice, in 1989 by the EU Scientific Committee for Food (SCF), and in 1990 by the Joint FAO/WHO Expert committee on Food Additives (JECFA) where an ADI of 0 to 0.01 mg/kg bw/day was established. Later, in 2011, EFSA concluded that it was not necessary to re-evaluate the ADI of erythrosine since the evidence available at the time did not provide a strong basis for toxicity. Moreover, the current levels of intake were under the stablished ADI, principally due to the low absorption that it presents [112]. E-127 is a food color with a combined maximum limit according to the European Regulation (CE) nº 1129/2011, and therefore the use conditions for these food additives in different food categories and the maximum level permitted were established in the regulations [48].

Betalains and especially betacyanins are natural pigments, which could replace the artificial pink colorants (Table 3). The food additive Betanin (E-162) is a natural food additive based on betalain content from beetroot red. its food colors were authorized at quantum satis according to the European Regulation nº 1129/2011 [48]. Several authors have worked on the development of a pink colorant from beetroot (*Beta vulgaris* L.). Lopez, et al. [113] have studied different conditions for the extraction of betalains through pulsed electric fields (PEFs) followed by a mechanical pressing. Parameters such as the electrical field strength (0–9 kV/cm) and the number of pulses (5–100) were evaluated for the PEF process. Meanwhile, the pH (3.0–6.5), temperature (10–60 °C) and the pressure of the press (0–14 kg/cm^2^) were evaluated for the mechanical pressing process. Through RSM, it was concluded that the best conditions were 5 pulses at 7 kV/cm for PEFs followed with mechanical pressing at 10 kg/cm^2^ for 35 min with an extracting medium with pH 3.5 and at 30 °C. In this way, a yield of 90% was achieved. On the other hand, Lombardelli et al. [114] determined the best enzymatic mix for the extraction of betalains from beetroot considering the polysaccharide composition of its cell wall (37% cellulose, 35% hemicellulose, and 28% pectin). Different parameters were evaluated, such as the total dosage of the acetate buffer containing the multi-component enzymatic mix (10–50 U/g), extraction temperature (25 or 45 °C) and time (0–5 h). Moreover, the pH (5.5) and the ratio s/L (1:15) were maintained constant. It was determined that 25 U/g of total dose enzymatic mix, 25 °C and 240 min were the best conditions, allowing an extraction yield of 11.37 mg/mL U to be obtained.

Although beetroot (*Beta vulgaris* L.) is the most explored natural source of betalains, there are other plants, which are highly interesting. *Gomphrena globosa* L. is a plant native to Latin America, whose flowers stand out for their betacyanin content, and which could be a remarkable source of these pigments. For this purpose, the bracts and bracelets of *Gomphrena globosa* L. were extracted by Roriz et al. [115] through maceration with different condition of time, temperature, water-ethanol proportion and s/L ratio. The betacyanin content (HPLC-PDA-MS/ESI) and the color intensity of each extraction were analyzed by response surface methodology (RSM) evidencing that the best extraction conditions were 0% of ethanol, 5 g/L of s/L ratio and 25 °C for 165 min, generating a red extract with a betacyanin content of 45 mg/g. In another study, the same authors aimed to evaluate the association of microwave (MAE)- and ultrasound-assisted extraction (UAE) processes, using RSM to obtain and identify the best conditions for the extraction of betacyanins from *G. globosa*. Their results showed that MAE and UAE are effective extraction techniques that were demonstrated to be useful for betacyanin extraction, with the advantage of decreasing the extraction time. The optimal processing conditions for MAE (8 min, 60 °C; 0% ethanol content; and solid/liquid ratio of 5 g/L) provided an extraction yield of 39.6 mg/g (dw). In the case of UAE, the conditions that provided the highest yield (46.9 mg/g) were 22 min; 500 W; 0% ethanol content; and solid/liquid ratio of 5 g/L. These findings indicated that UAE is a more suitable technique to obtain betacyanins from *G. globosa* [116].

Sivakumar et al. [86] evaluated the use of ultrasound extraction compared with magnetic stirring for obtaining extracts rich in anthocyanins from the 4 o’clock plant’s (*Mirabilis jalpa*) flowers and cocks comb (*Celosia cristata)* flowers. In the case of the 4 o’clock flowers, using ultrasound at 80 W for 3 h at 45 °C, an increase of 44.4% of extraction yield was generated. On the other hand, the same ultrasound conditions generated an increase in the yield of extraction of 14.3% for cocks comb flowers.

Cejudo-Bastante et al. [117] extracted betalains from Red prickly pear (*Opuntia dillenii* (Ker Gawl.) Haw.) through maceration for 24 h at 10 °C with methanol:water (60:40). The extract stability under different conditions of temperature (4, 20, 80 °C) and pH (4, 5, 6) was evaluated for 12 days. It was reported that at room temperature and acid pH, the extract changed to a yellowish color, but, at 4 °C, any significant change in the color was evidenced despite the pH, maintaining the initial red color and the betalain content.

In 2016, Otalora et al. [118] generated a purple extract from cactus (*Opuntia ficus-indica* L. Mill.) fruit pulp. First, the fruits were crushed in a homogenizer at maximum speed and room temperature for 5 min, and the betalains present in the pulp were extracted by maceration for 5 min with 0.1 M phosphate and 0.05 M citric acid pH 5 buffer as extraction solvent. Then, the extract was microencapsulated by ionic gelation with two different encapsulants, calcium alginate and a combination of calcium alginate and bovine serum albumin. After 25 days of storage, non-significant differences in the betalain retention from the samples with both encapsulants were evidenced. However, they increased the betalain retention in comparison with betalain extract without encapsulant agent. In addition, it was evident that the higher the moisture, the quicker the degradation of the colorant. Overall, they concluded that the best storage conditions were to encapsulate with calcium alginate at 34.6 RH at 25 °C, generating a betalain retention of 48.8%.

A juice from the complete pear fruit from another cactus species (*Opuntia stricta* (Haw.) Haw.) was provided by Obon et al. [119] using homogenization and centrifugation. The supernatant was spray-dried with glucose syrup as a drying aid and different levels of temperature, liquid feed rate and spray flow rate. Then, it was concluded that 160 °C, 0.72 L/h and 0.47 m^3^/h, were the optimum conditions for spray drying. After one month of storage, the solid colorant (initial color parameters: *L**: 13.99, *a**: 7.98, *b**: 0.57) maintained 98% of the color strength, evidencing a remarkable stability. A few years later, Koubaa et al. [120] compared the extraction through electric fields (PEFs) and ultrasound of the peel and pulp by-products. The ultrasound extraction was performed at 400 W and 100% of amplitude for 5, 10 and 15 min. The PEFs was carried out at voltages of 8, 13.3 and 20 kV/cm and from 50 to 300 pulses. In the case of ultrasound, 15 min was determined as the best time (50 mg/100 g fruit, fw) and 50 pulses at 20 kV/cm the best conditions for PEFs (60.0 mg/100 g, fw). Both methodologies allowed an improvement in yield extraction, however, the authors recommended the use of PEFs since they expend less energy, therefore, they are more sustainable.

In 2013, Sanchez-Gonzalez et al. [121], also worked with prickly pear fruit from another cactus species (*Opuntia joconostle* F.A.C. Weber), with different timings (10–30 min), temperatures (5–30 °C) and solvents (water, methanol-water, 4–80%, *v*/*v*; and ethanol-water (4–80%, *v*/*v*) tested for the extraction of the betalains of the fruit pulp. Through RSM, it was determined that the best extraction conditions were 15 °C for 10 min with methanol-water (20:80) as the solvent, which allowed 92 mg of betacyanin per 100 g of fruit to be obtained. Moreover, betalain, isobetalain, betanidin and isobetanidin were the main betalains identified in the extract through HPLC.

De Lima et al. [98] generated a red purple colorant from pitaya (*Hylocereus polyrhizus* (F.A.C. Weber) Britton & Rose). In addition, its stability was tested. The betalain extract was produced through maceration at 4 °C for 45 min at 150 rpm and an s/L ratio of 2 mg/mL, and through UPLC-Q-TOF-MS, the presence of some betalains as phyllocactins isomers, betanin and 6′-malonyl-2′-decarboxy-betanine isomers was identified. The color stability of the extract was tested across twelve months, evidencing non-significant changes in the CIELAB parameters (*L**: 0.53, *a**: 1.24, *b**: 0.34).

### 3.3. Source of Natural Yellow-Orange Colorants as a Potential Replacers of Artificial Colorants

Yellow 5 or tartrazine (E102) and yellow 6 or sunset yellow FCF (E110) are the most used colorants in the industry for the generation of yellow and orange color. In 2014, in the latest evaluation made of sunset yellow by EFSA, the temporary ADI of 1 mg/kg bw/day was replaced for the new ADI of 4 mg/ kg bw/day with assurances that the reported use levels are below this ADI and, therefore, its consumption will not represent any risk [122]. In 2009, on the other hand, the EFSA concluded that the database referring to adverse effects generated by tartrazine was insufficient, and therefore, a revision of the ADI was not necessary, maintaining in that way the ADI of 0–7.5 mg/kg bw/day. However, in the same scientific opinion, it was mentioned that tartrazine can generate intolerance reactions in part of the population and sensitive individuals can react to it within the ADI [123].

Currently, saffron (E164), beta-carotene (natural and synthetic), lycopene, tomato extract or concentrate (160) and turmeric or turmeric oleoresin (E100) are the colorants from natural sources accepted by the FDA and EFSA [124].

The food additive, Carotene (E-160a), is a natural food that can be extracted from natural strains of edible plants such as carrots and palm fruit oils. It is classified as a food color authorized at *quantum satis* according to the European Regulation nº 1129/2011 [48]. However, the mentioned regulation establishes a maximum level of 20 mg E-160a per kg only in the case of sausages, pâtés and terrines.

The majority of the research regarding carotenoids concerns its extraction and stabilization in order for it to be used as a functional ingredient. However, in recent years, research into the use of carotenoids as colorants has increased (Table 4). Sivakumar et al. [86] compared the extraction of marigold (*Tagetes erecta* L.) flowers through ultrasound and magnetic stirring; it was determined that ultrasound for 3 h at 80 W and 45 °C represent the best conditions for generating an extract rich in carotenoids with an improvement in the extract yield of 100%, obtaining an orange extract.

Coelho et al. [125] evidenced a faster and cheaper extraction of carotenoids from red cashew apple (*Anacardium occidentale* L.) through ultrasonic-assisted extraction than through conventional extraction. UAE was performed under conditions of 19 min of sonication and a mixture of 44% acetone and 56% methanol. From 153 mg of sample, 154.9 µg of carotenoid per gram of orange extract was obtained.

Palm press fiber is one of the highest by-products from the vegetal oil production; however, it presents its high amounts of carotenoids amongst other bioactive compounds. As a result, different efforts have been made for the extraction of these compounds, such as Dal prá et al. [126] who increased the extraction yield (4.55 wt%) by three times using supercritical CO2 at 60 °C and 25 MPa in comparison with extraction with liquefied petroleum gas. Later, Alvarega et al. [127] improved the extraction yield through the use of a mixture of hydrocarbons and alcohols in batch or column extraction obtaining from 1790 to 2539 mg of ß-carotene/kg of palm-pressed fiber.

The peel from Gac fruit (*Momordica cochinchinensis* Spreng.) is another by-product with high amount of carotenoids. In the latest studies, it was determined, through RSM, that 150 min, 40.7 °C, 80 mL/g as the solid–liquid ratio and ethyl acetate as the extraction solvent, were the optimal conditions through maceration, showing an extraction yield of 271 mg per 100 g, dw [128]. Later, a higher extraction yield with UAE was evidenced, rather than using microwave-assisted extraction, under conditions of 200 W and 80 min with ethyl acetate the as solvent, obtaining a total of 268 mg carotenoid yield per 100 g, dw [129].

Ripe bitter melon pericarp has been also studied for the extraction of carotenoids, such as Patel et al. [130] who treated ripe bitter melon pericarp with enzymes (load of 167 U/g) followed by extraction through supercritical fluid extraction, and obtained an extraction yield of 90.12% of ß-carotene under conditions of 390 bar of pressure, 35 mL/min of flow rate, 70 °C of temperature for 190 min. Moreover, a stability of 70% of retention was estimated until 2.27 months at 10 °C and up to 3.21 months at 5 °C.

**Table 4 foods-12-04102-t004:** Natural sources of yellow–orange colorants (carotenoids and betaxanthins), extraction conditions.

Pigment	Natural Source	Pigment Content in Extract	Extraction Conditions	Reference
Yellow–orange color				
Carotenoids	Red cashew apple (*Anacardium Occidentale* L.)	154.9 µg of carotenoid/gram of orange extract	Ultrasound-assisted extraction compared with conventional extraction: - solvents: 25–100% of acetone, ethanol, petroleum ether, methanol - time: 20 min - mechanical shaking: 290 rpm Optimal conditions for ultrasonic-assisted extraction: 19 min of sonication and a mixture of 44% acetone and 56% methanol.	[125]
	Palm (*Elaeis guineensis* Jacq.) press fiber	1140 mg β-carotene/100 g	Supercritical CO_2_ extraction: - temperature: 20–60 °C - pressure: 120–125 MPa Compressed liquefied petroleum gas (LPG): - temperature 20–40 °C - pressure: 0.5–2.5 MPa The higher amount of carotenoids (β-carotene) was obtained using LPG	[126]
		253.9 mg of ß-carotene/100 g	Cold extraction - solvents: ethanol, isopropanol, hexane, cyclohexane, heptane - s/L ratio: 1:5 - time: 8 h The highest carotenoid content was obtained with the use of hexane	[127]
	Gac fruit (*Momordica cochinchinensis* Spreng.) peel	271 mg per 100 g, dw	Extraction through maceration: - solvents: acetone, ethanol, ethyl acetate, hexane - time: 90–150 min - temperature: 30–50 °C - s/L ratio: 10–80 mL/g Optimal conditions were with ethyl acetate at a solid–liquid ratio of 80 mL/g and 40.7 °C for 150 min	[128]
		268 mg carotenoid/ 100 g, dw	Ultrasound-assisted extraction: - power: 150, 200 and 250 W - frequency: 43.2 kHz - temperature: 20 °C Microwave-assisted extraction: - power: 120, 240 and 360 W - temperature: until reached 60 °C Higher yield of carotenoids was obtained with ultrasound-assisted extraction under conditions of 200 W and 80 min with ethyl acetate as solvent.	[129]
	Strawberry tree (*Arbutus unedo* L.)	0.808 mg ß-carotene/100 g, fw	Maceration extraction: - solvent: hexane/acetone/ethanol 50:25:25 - magnetic stirring: 30 min	[131]
	Ripe bitter melon (*Momordica charantia* L.) pericarp	85.54 mg ß-carotene/100 g	Enzymatic treatment (load of 167 U/g) followed by supercritical fluid extraction: - pressure: 150–450 bar - CO_2_ flow rate: 15–55 mL/min - temperature: 50–90 °C - time: 45–225 min The optimal conditions for supercritical fluid extraction were 390 bar and 35 mL/min at 70 °C for 190 min	[130]
	Mango (*Mangifera indica* L.) peel	1.9 mg all-trans-ß-carotene/g, dw	Supercritical fluid extraction with CO_2_: - temperature: 40–60 °C - pressure: 25–35 MPa - ethanol as co-solvent: 5–15%, *w*/*w* The optimal extraction conditions for the highest yields of carotenoids were 25.0 MPa, 60 °C and 15% *w*/*w* ethanol	[132]
		5.6 mg of ß-carotene/g, dw	Supercritical CO_2_ extraction, followed by pressurized ethanol, both extractions methods at: - pressure: 30 MPa - temperature: 40 °C - ratio s/L: 3 g/10 mL A higher concentration of carotenoids was extracted with supercritical CO_2_ extraction	[133]
	Rowanberry fruit (*Sorbus aucuparia* L.)	19.14 mg carotenoids/g of extract	Consecutive extraction with supercritical CO_2_ followed by pressurized liquid extraction: Supercritical fluid extraction: - pressure: 25–45 MPa - temperature: 40–60 °C - flow rate of CO_2_: 2 SL/min - time: 180 min Pressurized liquid extraction: - temperature: 70 °C - pressure: 10.3 MPa - time: 15 min (3 cycles of 5 min of static extraction) The optimal extraction conditions with supercritical CO_2_, determined through RSM, were 45 MPa and 60 °C for 180 min.	[134]
	Persimmon fruits (*Diospyros kaki* L.)	15.46, 16.81, and 33.23 µg/g (dw) for all-trans-lutein, all-trans-zeaxanthin, all-trans-ß-cryptoxanthin, respectively	Extraction with supercritical CO_2_ and ethanol as cosolvent: - temperature: 40–60 °C - pressure: 100–300 bars - ethanol %: 5–25 - CO_2_ flow rate: 1–3 mL/min - time: 30–100 min The best conditions were 300 bars, 60 °C, 25% (*w*/*w*) ethanol at a flow rate of 3 mL/min flow for 30 min	[135]
		11.19 µg of all-trans-ß-carotene/g, dw	Supercritical CO_2_ extraction: - cosolvent: ethanol 25% (*w*/*w*) - pressure: 100 bars - temperature: 40 °C - flow: 1 mL/min - time: 30 min	[135]
		35.48–75.84 µg of carotenoids/100 g, fw	Ultrasound-assisted extraction: - s/L ratio: 2 g/50 mL - solvent: hexane:acetone:ethanol (50:25:25 *v*/*v*/*v*) - time: 10 min	[136]
betaxanthins	Beetroot (*Beta vulgaris* L.)	11.37 mg betaxanthins/L/U	Enzymatic extraction, including the evaluation of the following parameters: - total dosage of the acetate buffer containing the multi-component enzymatic mix: 10–50 U/g - temperature: 25 or 45 °C - time: 0–5 h - pH: 5.5 - ratio s/L: 1/15 It was determined that 25 U/g of total dose enzymatic mix, 25 °C and 240 min were the best conditions	[110]
	Cactus fruit (*Opuntia ficus indica*)	32.3–72.4 mg of betaxanthins/kg of juice	Maceration extraction: - solvent: methanol - s/L ratio: 1:5 *w*/*v* - time: 1 min - magnetic stirring The betaxanthins were more stable at pH 3.5 with ascorbic acid	[137]
	Cactus fruit (*Opuntia ficus indica*)	27.5 mg indicaxanthin/100 g	Extraction by homogenization with ultraturrax followed by centrifugation; the supernatant was determined as an extract rich in betaxanthin	[138]
	Pitahaya fruit peel (*Hylocereus megalanthus* (K.Schum. ex Vaupel))	0.1058 mg betaxanthin/g of sample	Maceration extraction: - solvent: ethanol/water (50/50 *v*/*v*) - s/L ratio: 10 g/300 mL - stirring: 700 rpm - temperature: room temperature - time: 5 h	[139]

s/l ratio: solid/liquid ratio.

Mango peel (*Mangifera indica* L.) is a by-product from the mango processing industry and presents a considerable quantity of carotenoids, principally *trans*-ß-carotene. An extraction yield of 1.9 mg all-*trans*-ß-carotene per gram of dried mango was obtained through supercritical fluid extraction with 25.0 MPa, 60 °C and 15% *w*/*w* ethanol as the extraction conditions [132]. In addition, the use of consecutive extraction methods, starting with supercritical CO_2_ extraction, followed by pressurized ethanol, both extraction methods at 30 MPa and 40 °C, showed a yield extraction of 5.6 mg of ß-carotene per g of dried mango peel [133].

Rowanberry fruit (*Sorbus aucuparia* L.) is another fruit that has been recently studied for its high content in bioactive compounds such as carotenoids, present in an amount around 78.91 mg/100 g of extract. A consecutive extraction with supercritical CO_2_ followed by pressurized solvent, evidencing through RSM that the optimal conditions were 45 MPa and 60 °C for 180 min, recovering 49.7% of the quantity of carotenoids present in the fruit, generating an orange extract [134].

Another fruit rich in carotenoids is persimmon (*Diospyros kaki* L.), and one of the latest works had determined, through RSM, the best conditions for extraction with supercritical CO2 and ethanol as the co-solvent of xanthophylls and all-trans-ß-carotene; for the first one, the best conditions were 300 bars, 60 °C, 25% (*w*/*w*) ethanol and 3 mL/min flow for 30 min obtaining yields of 15.46, 16.81 and 33.23 µg/g of dried fruit for all-*trans*-lutein, all-*trans*-zeaxanthin, and all-*trans*-ß-cryptoxanthin, respectively. For all-trans-ß-carotene, 100 bars, 40 °C, 25% (*w*/*w*) ethanol and 1 mL/min of flow for 30 min were the optimal conditions, showing an extraction yield of 11.19 µg/g of dried fruit [135].

The principal focus of the generation of natural colorants has been the individual extraction of different matrices; however, there have been a few studies where the goal was to achieve co-pigmentation through obtaining one extract derivate from two matrices. This is the case of Ciurlia et al. [140], who generated a lycopene oil extract from pulp and skin-dried tomato (*Lycopersicum esculentum* L.) and roasted hazelnuts (*Corylus avellana* L.). Through supercritical carbon dioxide co-extraction at 60 °C, 400 bar and 10 kg CO_2_/h of flow rate, the oil from hazelnuts and the lycopene from tomato were extracted simultaneously, allowing for an orange reddish oil colorant to be obtained with extraction yields of 72.5% for lycopene and 80% for hazelnut oil.

### 3.4. Source of Natural Green Colorants as a Potential Replacers of Artificial Colorants

The colorant normally use in United States by industry for obtaining green color is Fast green FCF (FD&C green No. 3); however, this colorant is banned in Europe and the artificial colorant authorized is Green S (E142), which in turn is banned in the United States. Nonetheless, in several cases, industry has obtained green colorants through mixing blue and yellow colorants, normally brilliant blue (E133) combined with tartrazine (E102) [141].

Green S colorant has an ADI of 5 mg/kg bw/day, defined in 1984 by the Scientific Committee on Food. In 2010, the last scientific opinion in the re-evaluation of green S was published, where it was concluded that a re-evaluation of the ADI was not necessary since it presented negative results for carcinogenicity; however, in the same scientific opinion it is mentioned that the estimated intake for children at the high percentiles is above the ADI [142].

Currently, the use is also allowed of two natural colorants, chlorophyll (E140i) and chlorophyllin (E140ii), oil-soluble pigments, and their corresponding homologs cu-chlorophyll (141i) and cu-chlorophyllin (E141ii), water-soluble pigments, resulting from the addition of copper, which improves their stability avoiding their change in color from green to brown [143]. Chlorophylls are principally extracted from grass, alfalfa, nettles, and spinach. Presently, there is no other type of colorant compound; nevertheless, there are several publications regarding new sources of chlorophylls, or new and more efficient ways of their extraction from the known sources [38].

The food additives, chlorophyll and chlorophyllin (E-140), are natural dietary constituents, which are present at relatively high concentrations in several foods. Plants commonly used for the extraction of chlorophylls include grass (*Festuca* spp.), alfalfa (*Medicago sativa* L.), nettle (*Urtica* spp.) and spinach (*Spinacia oleracea* L.) [39]. As food additives, chlorophylls and chlorophyllins are classified as food colors authorized at quantum satis according to the European Regulation nº 1129/2011 [48]. In addition, the EFSA Panel on Food Additives and Nutrient Sources added to Food (ANS) reported that the exposure resulting from the use of chlorophylls (E 140) as food additives is lower than the exposure from the regular diet, concluding that, at the reported use levels, chlorophylls (E 140) are not of safety concern as regards their use as food additives [39].

Table 5 shows the latest studies regarding natural sources of chlorophylls. Custard apple leaves (*Annona squamosa* L.) have been intensively studied in recent years due to their biological activities as an anticancer, antidiabetic, and antioxidant agent among others [144]. Moreover, some authors focus has been on their high amount of chlorophylls, such as Shiekh et al. [145], who concluded that, even though the use of pulsed electric fields at 6 kV/cm electric field strength, 300 pulses and 142 kJ/kg of specific energy for 5 min improved some bioactivities, it also led to the decrease in chlorophylls, evidencing a better extraction of chlorophylls without the use of pulsed electric fields (PEFs; 1.38 mg of total chlorophylls/g) than with PEFs (0.35 mg of chlorophylls/ g).

On the other hand, in one of the latest studies on broccoli (*Brassica oleracea* var. *italica*), it was determined that the use of high-intensity pulsed electric fields, at 26.35 kV/cm for 1.235 µs in bipolar mode, could be one of the best processes for the stability of the color properties in broccoli juice, mostly due to the reduction in the enzyme activity, showing a maximum ΔE* of 2.11 [146].

In the same way, a better stability of chlorophylls from spinach (*Spinacia oleracea* L.) was evidenced after the treatment with pulsed electric fields at 26 kV/cm of voltage, 35 °C and a pulse width of 20 µs, obtaining 20.75 µg of total chlorophyll/ mL of juice. It was expected that the higher voltage, the higher the stability of the chlorophylls would be [147].

*Centella asiatica* L. leaves are another source of chlorophylls. Ngamwonglumlert et al. [148] performed an ultrasonic-assisted extraction of the leaves after a treatment with zinc or copper for the generation of complexes, the untreated extract presented a light green color (*L**: 40.17, *a**: −11.13, *h**: 116.49), while the zinc and the copper-treated extracts presented a yellow-green color (L*: 21.43, *a**: −6.95, *h**: 113.10). In addition, a better stability against acid pH and heat was evidenced in the extract treated with metals.

Apart from the stabilization of chlorophylls through copper, a few authors have studied stabilization through encapsulation. An increase in the stability (994.7–97.5%) of an extract rich in chlorophylls from spinach, due to encapsulation with maltodextrin, was evidenced after ten days of storage at 4, 20 and 40 °C. Maltodextrin was the better encapsulation agent, showing an encapsulation efficiency of 77.19%, and a chlorophyll content of 46.78 µg/ g of dry powder with a strong green color (*L**: 26.22, *a**: −16.72, *b**: 19.64) [149].

**Table 5 foods-12-04102-t005:** Natural sources of green colorants, extractions conditions and color parameters.

Pigment	Natural Source	Pigment Content	Extraction Conditions	Optimal Extraction Conditions/Color Parameters	Reference
Green color					
Chlorophylls	Custard apple (*Annona Squamosa* L.) leaves	1.38 mg of total chlorophylls/g (without the use of PEFs), and 0.35 mg of chlorophylls/ g (by using PEFs)	Pulsed electric fields extraction: - electric field strength: 6 kV/cm, - pulses: 300 - specific energy: 142 kJ/kg - time: 5 min	A better extraction of chlorophylls was evidenced without the use of pulsed electric fields than with it.	[145]
	Broccoli (*Brassica oleracea* L. var. italica)	-	High-intensity pulsed electric fields: - time: 500–2000 µs - electric field strength: 15–35 kV/cm - polarity: monopolar or bipolar - pulse width: 4 µs - frequency: 100 Hz	The optimal conditions, according to the color parameters (ΔE* of 2.11), were 26.35 kV/cm for 1.235 µs in bipolar mode.	[146]
	Spinach (*Spinacia oleracea* L.)	20.75 µg of total chlorophyll/ mL of juice	Maceration extraction followed by pulsed electric fields treatment Maceration extraction (consecutive extraction): - solvent: absolute ethanol - ratio s/L: 200 g/ 400 mL - time: 30 min - centrifugation: 500 g for 10 min Pulsed electric fields treatment: - electric field intensity: 0–26.7 kV/cm - temperature: 20–45 °C	The best pulsed electric field conditions were 26 kV/cm of voltage, 35 °C and pulses width of 20 µs.	[147]
		46.78 µg/ g of dry powder	Chlorophyll extract was dispersed in MCT oil to a concentration of 0.01 g/mL followed by stabilization by microencapsulation: - wall materials: maltodextrin, gum Arabic - intel air temperature: 145 °C - outlet temperature: 95 °C - pump speed: 1.5 mL/min	The best wall material was maltodextrin with color parameters of: *L**: 26.22, *a**: −16.72, *b**: 19.64, *h*: 131.43, *C**: 25.78.	[149]
	*Centella asiatica* L. leaves	-	Ultrasonic-assisted extraction of the leaves after a treatment with zinc or copper for the generation of complexes: Ultrasonic-assisted extraction: - ratio s/L: 10 g/200 mL - time: 10 min - frequency: 37 kHz - power: 500 W	The untreated extract presented a light green color (*L**: 40.17, *a**: −11.13, *h*: 116.49), while the zinc and the copper-treated extracts presented a yellow–green color (*L**: 21.43, *a**: −6.95, *h*: 113.10). A better stability against acid pH and heat was evidenced in the extract treated with metals.	[148]

*L**: lightness; *a**: red/green coordinate; *b**: yellow/blue coordinate; *h*: hue; *C** chroma; s/l ratio: solid/liquid ratio.

The nanoencapsulation of an extract rich in anthocyanins from spinach with zein, casein and whey protein was tested; all the carrier agents improved the stability, and the extract with casein presented as a light green (*L**: 46.99, *a**: −7.33, *h**: 120.11); the extract with whey protein presented as a military green (*L**: 38.53, *a**: −0.72, *h**: 100.62); and the extract with zein presented as a brownish–green color (*L**: 36.40, *a**: −0.50, *h**: 117.51). However, zein showed the highest chlorophyll retention in the presence of light (65.5%) and acid pH (97.8%) [150].

### 3.5. Source of Natural Blue Colorants as a Potential Replacement for Artificial Colorants

Regarding artificial blue colorants, brilliant blue FCF (E133, FD&C Blue No. 1) and indigo carmine/indigotine (E132, FD&C blue No. 2) are approved as food colorants in the European Union and the United States [151].

According to the European Regulation nº 1129/2011 [48], brilliant blue (E 133), and indigotine/indigo carmine (E 132) are classified as food colors with a combined maximum limit. For brilliant blue (E 133), the maximum levels approved are 20 mg/kg only for processed mushy and garden peas (canned); 200 mg/kg only for preserves of red fruit; and quantum satis in the case of unprocessed meat other than meat preparations as defined by Regulation (EC) nº 853/2004, only for the purpose of health marking. The Panel on Food Additives and Nutrient Sources added to Food provided a scientific opinion re-evaluating the safety of brilliant blue FCF (E 133) and established an ADI for this food colorant of 6 mg/kg body weight/day [152]. Recently, the EFSA Panel on Food Additives and Flavourings (FAF) has confirmed the ADI of 5 mg/kg body weight/ day for indigo carmine (E 132) disodium salts, concluding that there is no safety concern for the use of this food colorant [153]

Among the natural sources that allow blue pigments to be obtained, which could be used by the food industry as colorants, anthocyanins found in blue color flowers and fruits are highlighted (Table 6) [22]. As has been previously explained, the color of the anthocyanins depends on the pH. In this sense, these compounds exhibit a blue color at pH 7–8 [101]. Butterfly pea (*Clitoria ternatea* L.) flowers are used for culinary purposes in the southern Asian region. In particular, the anthocyanins found in the petals of this plant could be employed as a blue food colorant [151]. For this purpose, Maneechot et al. [154] explored the suitable proportion of butterfly pea petals to use in traditional rice cooking. These authors carried out an aqueous extraction with different proportions of dried butterfly pea petals and they observed that the final product was better accepted by consumers when the incorporated extract of *Clitoria ternatea* was 0.6% (*w*/*v*). The cooked rice samples that contained the extract at 0.6% had a total anthocyanin content of 0.15 mg of cyanidin-3-glucoside equivalents per 100 g (dw). The aqueous extract of butterfly pea has been also used as natural colorant in gummy candies, which were well-accepted by consumers [155]. The anthocyanins of the blue flowers of *Centaurea cyanus* L., also known as cornflower, have been also investigated. The hydromethanolic extracts (methanol/water: 80/20, *v*/*v*) showed a total anthocyanin content of 27 μg/g and it was concluded that these flowers had an interesting potential for use as a source of natural food colorants in the blue range [156].

Apart from anthocyanins, spirulina/phycocyanin could be used as a blue food colorant. Phycocyanin is a heat-labile protein complex extracted from *Spirulina platensis*, which can be found in bakery products and ice creams [22]. However, the culinary applications of phycocyanin are limited due to its sensitivity to heat treatment that causes its precipitation and significant decrease in the blue color [158]. Different eco-friendly techniques have been investigated to extract food blue colorants from the dry biomass of spirulina. These methods include the use of nanoparticles, polyethylene glycol and salts. It has been observed that among the stated methods, extraction using silver nanoparticles resulted in the highest yield (47.68 of C-phycocyanin and 34.18 mg of A-phycocyanin per gram, dw) and purity (0.98 and 0.33, for C-phycocyanin and A-phycocyanin, respectively) [157].

As evidenced above, there is a huge variety of plants, fruits, flowers and cereals, among others, that contain representative amounts of colorant compounds and therefore can be considered as sources of natural colorants; in addition, intensive efforts have been made in order to improve their extraction as well as their stability. However, it is important to mention that big differences were found between the matrices contemplated in this review, in terms of composition and behavior, in addition to the enormous differences between methodologies and determinations carried out in each study, which amplifies the possibilities for improvement and the obtention of better natural colorants but also makes comparisons not just between different matrices for obtaining similar colorant compounds but also between the same matrices challenging.

## 4. Incorporation of Natural Food Colorants in Food Matrix

Different studies report the novel use of natural food colorants in food product formulations in order to replace the use of synthetic ones. In this sense, Table 7 shows the most recent studies in this field.

### 4.1. Incorporation of Anthocyanins in Food Matrix

Different authors have incorporated extracts of anthocyanins in yogurt (Table 7). In a study performed by Nontasan et al. [109], a dark red extract rich in anthocyanins obtained from black rice bran was added to this matrix. Any significant change in the pH and total acidity in the yogurt was evidenced by the addition of the colorant. Moreover, the color changed in accordance with the amount of colorant added, pink with 0.2%, the least amount (*L**: 80.20, *C**: 9.67, h: 18.34) and purplish pink with 0.6%, the highest percentage added (*L**: 68.42, *C**: 15.23, h: 8.73). In addition, any change in the color of the yogurt was evidenced during 21 days of storage at 4 °C. Mourtzinos et al. [53] also worked on improving the color of yogurt by adding a red extract rich in anthocyanins from red onion by-products. It was added before and after the thermal treatment and then storage at 5 °C for four weeks. The color parameters as well as the amount of total anthocyanin content were measured at time zero (*L**: 75.27, *a**: 11.20, *b**: 9.55; 0.05 mg cya-3-gluE/g) and after 4 weeks of storage (*L**: 75.19, *a**: 11.35, *b**: 9.75; 0.039 mg cya-3-gluE/g), a slight decrease in the total anthocyanin content was evidenced; however, there was no significant change in the color parameters with an ΔΕ* under 0.04. An extract rich in anthocyanins from eggplant (*Solanum melongena* L.), free and encapsulated with gum Arabic, was added to yogurt in concentrations of 1.0, 1.5 and 2 g/100 mL of yogurt, and its stability was evaluated during 30 days; a total loss of the anthocyanins in the samples with free extract was evidenced at the end of the month, while in the samples with encapsulated extract, the stability of anthocyanins at the end of the month was from 85 to 98%. In addition, the sample with the best retention of color (98.83%) after the 30 days was the yogurt with 1% of encapsulated extract with *L** of 83.74, *a** of −3.96 and *b** of 8.18 as color parameters [159].

The suitability of onion skin extract for the development of bakery products was also evidenced with the addition of 1% and 3% of extract into crackers. The extracts were previously encapsulated with 1.5% of soy protein isolate, 3% of gum Arabic and 3% of CMC sodium carboxymethyl cellulose presenting color parameters of *L**: 22.56, *a**: 20.84 and *b**: −0.78. The sample with 3% of encapsulated extract presented a retention of anthocyanins of 85.5% after 28 days of storage. Moreover, in the hedonic sensory analysis, performed using untrained panelists, the same sample obtained the best score in all parameters, especially in terms of color and firmness [160].

Black carrot extract was tested in marmalades showing a loss from 79.2 to 89.5% of anthocyanins due to the thermal processing parameters of the product. In addition, the stability of the anthocyanins in the marmalades was evaluated for 20 weeks, evidencing a better conservation at 4 °C (53.4–81.0%) than at 20 °C (7.8–69.3%) [161].

On the other hand, some extracts rich in anthocyanins have been added to food products in order to improve bioactivity, such as antioxidant activity, which has led to products with new characteristics, as in the case of obtaining red beer through the addition of eggplant peel extract. The antioxidant activity measured through DPPH methodology increased from 77.7% to 85.82%; additionally, the antioxidant activity of the beer enriched with the extract presented a higher stability (up to 92%) during 21 days of storage. Moreover, the color passed from a light yellow (*L**: 90.64, *a**: −2.07, *b**: 18.47) to a reddish yellow (*L**: 74.67, *a**: 9.86, *b**: 21.76) [162].

### 4.2. Incorporation of Betalains in Food Matrix

A purple extract rich in betalains from cactus (*Opuntia ficus-indica* L. Mill.) fruit pulp, encapsulated with sodium alginate, was added to gummy candies. Previous to the addition, the encapsulated betalains were released in water by orbital shaking in order to guarantee a total incorporation in the matrix. The colorant was added in 5 different ratios (30, 40, 50, 60 and 70%), where the higher the amount of added colorant, the darker the color of the gummy candies. Moreover, a maximum of ΔE* of 4 was evidenced after 30 days of storage at 4 °C, evidencing the high stability of the color in the gummy candies [164].

De Lima et al. [98] incorporated a red-purple colorant in yogurt, rich in betalains extracted from pitaya (*Hylocereus polyrhizus* (F.A.C. Weber) Britton & Rose). In this work, a general characterization and sensory analysis were carried out. Six samples were evaluated, four with different percentages of the colorant (0.5, 1.0, 1.5 and 2%), and two commercial yogurts, one with beetroot colorant and the other with carmine colorant. The sensory analysis was performed with 51 untrained panelists, half women and from 18 to 55 years old. Four tests were made, general opinion with a right scale from 1 to 7, acceptance and preferences with a 9-point hedonic scale, intensity of flavor with a 9-cm non-structure scale and purchase intent with a 5-point structured scale. In the characterization, it was concluded that the colorant did not affect the yogurt in terms of pH, acidity, and content of soluble solids. Regarding the sensorial analysis, the yogurts with 0.5% and 2% of the colorant were statistically similar to the commercial yogurts with beetroot and carmine colorants, respectively. Finally, the yogurt with 2% of colorant was the best evaluated in all tests; therefore, it was categorized as a great possible red–purple natural colorant.

As mentioned in the previous sections, Roriz et al. [165] generated two solid extracts rich in betalains from *Gomphrena globose* L.; one was dried by lyophilization and the other by spray drying with 20% of maltodextrin. Then, these colorants were incorporated into cookies; additionally, cookies without colorant and with commercial colorant E162 were prepared as the control. The four cookie samples were stored for 1 month in the cold and analyzed every week. The addition of the colorant did not alter the physical or chemical characteristics of the cookies. In respect of the color, the samples with the addition of the lyophilized (*L**: 56.3, *a**: 22.1, *b**: 8.0) and spray-dried (*L**: 56.0, *a**: 25.5, *b**: 4.3) colorant showed a pink color, while the sample with the commercial (*L**: 60.7, *a**: 23.2, *b**: 18.1) colorant was closer to an orange tone. Across the month of storage, the variation in color was low, mostly due to the low presence of water and therefore the absence of oxidation. Roriz et al. [166] have also extracted betalains from the flowers of *Amaranthus caudatus* L. and from red-fleshed pitaya peels, and the obtained extract was incorporated into tagliatelle pasta and meringue cookies, respectively. These results showed that betalains can be used as natural pink colorants in both savory and sweet food products.

### 4.3. Incorporation of Carotenoids in Food Matrix

A red extract rich in carotenoids from tomato skin (particularly rich in lycopene) was added to butter (20 mg/kg), ice cream (70 mg/kg) and mayonnaise (50 mg/kg). The stability at 4 to 6 °C in the case of butter and mayonnaise and at −25 °C for ice cream was evaluated for four months. Moreover, a sensory evaluation through a 9-point Hedonic scale was carried out with a semi-trained panel; characteristics such as appearance, color, aroma, taste, texture and overall acceptability were evaluated. Regarding the color parameters, for butter at the initial time, color parameters of *L**: 48.78, *a**: 2.36 and *b**: 14.17 were obtained, while at four months of storage the values were *L**: 48.37, *a**: 2.25 and *b**: 14.18. Values of *L**: 84.41, *a**: 11.56 and *b**: 28.59 were obtained for initial storage and values of *L**: 83.97, *a**: 11.16 and *b**: 21.29 at four months of storage for ice cream and in the case of mayonnaise, values of *L**: 55.82, *a**: 10.32 and *b**: 19.95 were obtained at the initial time and *L**: 53.66, *a**: 9.64 and *b**: 19.10 at four months of storage. In view of the above, non-significant changes were evidenced in the color of the butter, ice cream or mayonnaise from time zero to four months. Regarding the sensory evaluation, in most of the parameters for the three products, a decrease in the score was evidenced in the fourth month of storage, except in the case of mayonnaise where significant changes were evident after two months. However, it was evident that color was the most stable parameter, especially in the ice cream where it did not present a significant change [163].

Rizk et al. [91] also added a red extract rich in lycopene from tomato peel into ice cream; however, it was added in different percentages (0, 1, 3, 4 and 5%), and no change in the functional properties or general composition (protein %, pH, specific gravity, total solid%, between others) was evidenced. However, the only parameter influenced was the freezing point, where the higher the amount of extract added, the lower the freezing point. In addition, a sensory evaluation of all the samples was carried out evaluating flavor, body, and texture, melting and color at preparation and after 30 days of storage. The ice cream prepared with 3% of extract obtained the highest score in all the parameters followed by 4%, 2%, 1% and 5%. Moreover, the same pattern was evidenced after 30 days of storage and there was no significant change along the time.

The addition of mango peel powder into different matrices was also studied, different percentages from 5 to 20% of mango peel powder were added to dough biscuits, the best contents of total dietary fiber (increase of 20.7%), polyphenols (4.50 mg/g of biscuit, 8-fold) and carotenoids (247 µg/g of biscuit, 14-fold) were achieved with the highest percentage of mango peel powder (20%); however, in the hedonic sensory evaluation, a better acceptability of the biscuits with 10% of mango peel powder was evidenced. In terms of color the higher the amount of mango peel powder added, the darker the yellow color of the biscuits (20%: *L**: 52.90, *a**: 7.71, *b**: 22.02) compared with the control (*L**: 64.87, *a**: 9.52, *b**: 26.26) [171]. Later, mango peel powder was also added to macaroni in three different percentages from 2.5 to 7.5%, and a similar pattern as the one seen with the biscuits was evidenced in the macaroni. With the highest addition of mango peel powder (7.5%), a higher increase in the total dietary fiber content was observed (17.8%), polyphenols (1.80 mg GAE/g of macaroni; 4-fold) and carotenoids (84 µg/ g of macaroni; 18-fold). Nevertheless, the best cooking quality and texture as well as the sensory evaluation were shown with the addition of 5% of mango peel powder [167].

### 4.4. Incorporation of Chlorophylls in Food Matrix

As was mentioned before, Ngamwonglumlert et al. [148] obtained an extract rich in chlorophylls from *Centella asiatica* L. and improve its stability by generating complex with zinc and copper. Later the stability of the extract was tested in syrup and bread, a change in color from green to yellowish was evidenced in the syrups with the untreated extract and after pasteurization, the color was totally yellow, however, the syrups with the extracts treated with zinc and copper, were stable and present minimum change in color, mostly an increase in the *h** value. In the case of the bread, the untreated extract as the one treated with zinc, presented a yellow-green color, while the bread with the extract treated with zinc presented a green color. Additionally, the stability of the products was evaluated during seven days, during which no significant change was evidenced.

*Nannochloropsis oculata* D.J. Hibberd free and encapsulated with maltodextrin was added into white chocolate in four different percentages from 0.25 to 0.75%. The samples with the encapsulated extract present a higher amount of chlorophylls (22.9 µg/g of chocolate for 0.75%) than the samples with the free extract (20.5 µg/ g of chocolate for 0.75%); however, the samples with free extract presented a better stability of the color over the 28 days that it was evaluated under accelerated conditions (25 °C and 70% of relative humidity). Nonetheless, it is important to highlight that all of the samples presented a ΔΕ* over 3.0, therefore, all of the samples presented visible changes. In addition, the sensory analysis, showed that in the samples with the highest amount of encapsulated extract the acceptability decrease, however, this problem was not evidenced with the use of the free extract [168].

Extracts rich in chlorophylls from *Isochrysis galbana* Pascher and *Nannochloropsis oculate* were added to chewing gum at 0.5 or 1%. The samples prepared with *N. coulata* presented a yellowish color; however, the samples with *I. galbana* presented a green color, even with the smallest amount. All the samples showed a decrease in hardness as well as in cohesiveness, which is advantageous for a chewing gum. It was also evidenced that the chlorophylls were not degraded by the production process, and this can be seen in the sensory analysis which showed a high acceptability of the samples without a significant difference between them [169].

Green yogurt has been another goal; therefore, spirulina powder was added in yogurt in different concentrations from 0.25 to 1%, this led to the improvement of mouth feel and apparent viscosity. The preparation with 0.25% was found to be the best amount since it was sufficient to accelerate the end of fermentation and most importantly, presented the best acceptability in the sensory analysis. Moreover, there was no significant change in the color after 28 days of storage at 4 °C [170].

## 5. Conclusions

The high demand for healthier food products has led to research into new sources of natural colorants, which do not present side effects in the same way as artificial colorants, and on the contrary, show biological activity which could lead to health benefits. Nowadays there are several natural sources of food colorants, which could be used as alternatives for synthetic colorants. These sources include fruits, vegetables, flowers and algae, among others, and anthocyanins, betalains, carotenoids, chlorophylls and phycocyanins are some of the natural pigments extracted from them. In the European Union, the EFSA evaluates all safety concerns related to food additives, including artificial and natural colorants, and it has been observed that in recent years there have been modifications related to the acceptable daily intake (ADI). This fact indicates that knowledge about food additives is continuously growing. However, it also highlights the lack of more in-depth and reliable studies regarding the side effects of the consumption of artificial colorants so that the responsible entities can take appropriate measures. Through this review, the variety of sources of natural colorants available to date has been evidenced, in addition to the existence of different methods for the extraction of their colorant compounds (such as maceration processes using different solvents, the application of supercritical fluids, processes assisted by enzymes, microwaves or ultrasound, the use of pulsed electric fields, or even employing nanoparticles, as well as the different possibilities for their stabilization such as microencapsulation, nanoencapsulation, co-pigmentation, among others). This wide range of possibilities, as well as the numerous natural sources that could be employed to extract colorants and the difference between the same matrices due to different locations or time of growing and cropping, among others, makes it difficult to obtain homogeneous results in terms of pigment concentration or the color characteristics of the extract. Moreover, it is important to mention the low stability presented by the natural colorants against light, pH, temperature and oxygen, among others, which has been one of the biggest disadvantages of these colorants. In the same way, it is important to highlight the great developments that have been made to solve these stability problems. Finally, this review also deepens knowledge of the application of natural food colorants showing that they can be incorporated into different food products, such as yogurt, bakery products, gummy candies, ice cream or pasta. To conclude, the above mentioned shows that more research is needed in order to improve the obtention and stability of natural colorants, so they can be more widely and easily used. Also, the food industry must take more account of the developments achieved by researchers and increase efforts in terms of the incorporation of natural colorants in different products for healthier food product development, without neglecting compliance with all the regulatory requirements established in the current legislation.

## Figures and Tables

**Figure 1 foods-12-04102-f001:**
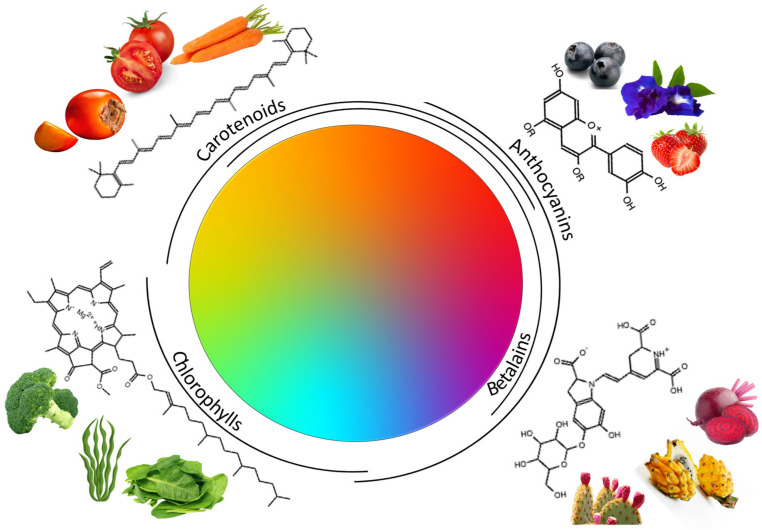
Principal colorant compounds found in natural sources.

**Table 1 foods-12-04102-t001:** Natural food colorants with approved use in Europe and United States, by EFSA and FDA, respectively [42,43].

Color	Origin	Scientific Name	Chemical Compound	Isolation	E Number	CFR Section
Blue	Butterfly pea flower extract	*Clitoria ternatea* L.	Anthocyanins	Aqueous extraction	NA	73.69
Blue-green	Spirulina extract	*Arthrospira platensis*	Phycocyanins		NA	73.530
Green	Edible plant material, grass, lucerne and nettle extract	*Gramineae*, *Medicago sativa* L., *Urtica dioica* L.	Chlorophylls	Solvent extraction	E140i Cl natural green 3; magnesium chlorophyll; magnesium phaeophytin	
Green-blue	Edible plant material, grass, lucerne and nettle extract	*Gramineae,* *Medicago sativa* L., *Urtica dioica* L.	Chlorophyllins	Solvent extract and saponification	E140ii Cl natural green 5; sodium chlorophyllin; potassium chlorophyllin	
Green-blue	Edible plant material, grass, lucerne and nettle extract	*Gramineae,* *Medicago sativa* L., *Urtica dioica* L.	Copper complex of chlorophylls	Solvent extraction and addition of a salt of copper	E141i Cl natural green 3; copper chlorophyll; copper phaeophytin	
Green-blue	EU: Edible plant material, grass, lucerne and nettle. US: alfalfa extract	US: *Medicago sativa* L.	Copper complex of chlorophylls	Solvent extract, saponification, and addition of a salt of copper	E141 (ii) CI Natural Green 5; Sodium Copper Chlorophyllin; Potassium Copper Chlorophyllin	73.125
Yellow	Turmeric extract	*Curcuma longa* L.	Curcumin	Solvent extraction	E100 Cl natural yellow 3, turmeric yellow, diferoyl methane	73.600
	Turmeric oleoresin	*Curcuma longa* L.	Curcumin	Extraction with exact solvents	NA	73.615
Yellow-orange	Edible plants, carrots, vegetable oils, grass, alfalfa, and nettle extract	*Daucus carota* L.	Carotenoid—Beta-carotene	Solvent extraction	E160a (ii) Cl food orange 5	73.95
Yellow-orange	Algae extract	*Dunaliella salina* Teod.	Carotenoid—Beta-carotene	Essential oil extraction	E160a (iv) Cl Food orange 5	
Yellow-brown	Edible fruits and plants, grass, lucerne and African marigold extract	African marigold: *Tagetes erecta* L.	Carotenoids–Lutein	Solvent extraction	E161b Lutein; mixed carotenoids; xanthophylls	
Red-brown	Annatto tree seeds outer coating extract	*Bixa orellana* L.	Carotenoid–bixin	Solvent extraction	E160b (i) Annatto; bixin; norbixin; Cl natural orange 4	73.30
	Annatto tree seeds outer coating extract	*Bixa orellana* L.	Carotenoid–bixin	Alkali extraction	E160b (ii) Cl natural orange 4	
	Annatto tree seeds outer coating extract	*Bixa orellana* L.	Carotenoid–bixin	Vegetal oil extraction	E160b (ii) Cl natural orange 4	
Yellow-orange	Carrot oil	*Daucus carota* L.	Carotenoid	Solvent extraction	NA	73.300
Yellow-orange	Dried stigma powder	*Crocus sativus* L.	Carotenoid–crocin	Dry grinding	NA	73.500
Red	Ground dried paprika	*Capsicum annuum* L.	Carotenoid–capsanthin and capsorubin	Dry grinding	NA	73.340
Red	Paprika oleoresin	*Capsicum annuum* L.	Carotenoid–capsanthin and capsorubin	Solvent extraction	E160c	73.345
Red	Red tomatoes extract	*Lycopersicon esculentum* L.	Lycopene	Solvent extraction	E160d (ii) Natural yellow 27	73.585
	Red beets roots extract	*Beta vulgaris* L. var. *Rubra*	Betalaine	Pressing, aqueous extraction or dehydrating	E162 Beet red	73.40
Red-purple	Grape or black carrot extract	*Vitis vinifera* L., *Daucus carota* L.	Anthocyanin	Aqueous extraction	163	73.169
	Grape skin extract	*Vitis vinifera* L.	Anthocyanin–enocianina	Aqueous extraction	163	73.170

E number: European number; CFR: Code of Federal Regulations; NA: not approved.

**Table 3 foods-12-04102-t003:** Natural sources of pink colorants (betacyanins), extraction conditions and color parameters.

Pigment	Natural Source	Pigment Content in Extract	Extraction Conditions	Reference
Pink color				
Betacyanins	Beetroot (*Beta vulgaris* L.)	-	Extraction through pulsed electric fields (PEFs) followed by a mechanical pressing. Parameters for the electric field: - strength: 0–9 kV/cm - number of pulses: 5–100 Parameters for the mechanical pressing process: - pH: 3.0–6.5 - temperature: 10–60 °C - pressure of press: 0–14 kg/cm^2^ The best conditions: 5 pulses at 7 kV/cm for PEFs followed by mechanical pressing at 10 kg/cm^2^ for 35 min with an extracting medium with pH 3.5 and at 30 °C.	[113]
		14.67 mg betacyanins/L/U	Enzymatic extraction, including the evaluation of the following parameters: - total dosage of the acetate buffer containing the multi-component enzymatic mix: 10–50 U/g - temperature: 25 or 45 °C - time: 0–5 h - pH: 5.5 - ratio s/L: 1/15 The best extraction conditions: 25 U/g of total dose enzymatic mix, 25 °C and 240 min were the best conditions.	[114]
	Bracts and bracelets (*Gomphrena globosa* L.)	45 mg/g, dw	Extraction through maceration with different conditions of time, temperature, water–ethanol proportion, and s/L ratio. The best extraction conditions: 0% of ethanol, 5 g/L of s/L ratio, and 25 °C for 165 min.	[115]
		39.6 mg/g, dw	Microwave-assisted extraction (MAE). The optimal processing conditions: 8 min; 60 °C; 0% ethanol content; and solid/liquid ratio of 5 g/L.	[116]
		46.9 mg/g, dw	Ultrasound-assisted extraction (UAE). The optimal processing conditions: 22 min; 500 W; 0% ethanol content; and solid/liquid ratio of 5 g/L.	[116]
	Red prickly pear (*Opuntia dillenii* (Ker Gawl.) Haw.)	-	Extraction through maceration for 24 h at 10 °C with methanol:water (60:40) as solvent. The best extraction conditions: At room temperature and acid pH the extract changed to a yellowish color; however, at 4 °C, any significant change in the color parameters was evidenced despite the pH value, maintaining the initial red color and the betalain content.	[117]
	Cactus fruit (*Opuntia ficus indica* L. Mill) pulp	-	Extraction by maceration for 5 min with 0.1 M phosphate and 0.05 M citric acid pH 5 buffer as extraction solvent. The best storage conditions: to encapsulate the extract with calcium alginate at 34.6 RH at 25 °C.	[118]
	Cactus fruit (*Opuntia stricta* (Haw.) Haw.)	357 mg betain /100 g powder	Spray dried. The optimum conditions of spray drying were 160 °C, 0.72 L/h and 0.47 m^3^/h.	[119]
	Cactus fruit (*Opuntia stricta* (Haw.) Haw.) Peels and pulp by-products	60 mg/100 g, fw	Extraction through electric fields (PEFs) (voltages of 8, 13.3 and 20 kV/cm and from 50 to 300 pulses). The optimal conditions: 50 pulses at 20 kV/cm were the best conditions	[120]
		50 mg/100 g fruit, fw	Extraction through ultrasound (400 W and 100% of amplitude for 5, 10 and 15 min). The optimal conditions: 15 min was determined as the best time.	[120]
	Prickly pear fruit (*Opuntia Joconostle* Cv)	92 mg of betacyanin/100 g of fruit	Different extraction parameters: - time: 10–30 min - temperature: 5–30 °C - solvents: water; methanol-water (4–80%, *v*/*v*); and ethanol-water (4–80%, *v*/*v*). The best extraction conditions: 15 °C for 10 min with methanol-water (20:80) as solvent	[121]
	Pitaya (*Hylocereus polyrhizus* (F.A.C. Weber) Britton & Rose)	-	Extraction through maceration at 4 °C for 45 min at 150 rpm and a s/L ratio of 2 mg/mL.	[98]

*L**: lightness; *a**: red/green coordinate; *b**: yellow/blue coordinate; *h*: hue; *C**: chroma; s/l ratio: solid/liquid ratio.

**Table 6 foods-12-04102-t006:** Natural sources of blue colorants, *extraction conditions*.

Pigment	Natural Source	Pigment Content	Extraction Conditions	Reference
Blue color				
Anthocyanins	Butterfly pea (*Clitoria Ternatea* L.) flowers	4.48 mg cyanidin-3-glucoside equivalent/100 g of powder dw	Butterfly pea petals were cleaned, dried and ground to a fine powder	[154]
	Flowers of *Centaurea cyanus* L.	27 µg of anthocyanins/g of extract	Maceration extraction: - solvent: methanol/water: 80:20, *v*/*v* - temperature: 25 °C - agitation: 150 rpm - time: 1 h	[156]
Phycocyanin	Spirulina (*Arthrospira platensis*)	47.68 of C-phycocyanin/g (dw) 34.18 mg of A-phycocyanin/g (dw).	Extraction by using silver nanoparticles: - nanoparticles: silver, gold and aluminum oxide - nanoparticle concentration: 0–25 µg/mL - solvent: 0.1 M phosphate buffer and distilled water - salt concentration: 0–25 mg/mL - PEG concentration: 0–30% *w*/*w* - time: 120–200 min - temperature: 2–35 °C The best conditions were 10 µg/mL of silver nanoparticles, 3 mg/mL of salt, 10% of PEG, phosphate buffer at 35 °C for 160 min	[157]

**Table 7 foods-12-04102-t007:** Recent studies of natural food colorants incorporated in different food matrices.

Food Color	Natural Colorant	Food Matrix	Natural Sources	Main Food Characteristics: Color and Stability	Reference
Red-purple	Anthocyanins	Yogurt	Black rice bran extract	Pink color with 0.2% (*L**: 80.20, *C**: 9.67, h: 18.34) and purplish pink with 0.6%, the highest percentage added (*L**: 68.42, *C**: 15.23, h: 8.73)	[109]
		Yogurt	Red onion by-products	Time zero: 0.05 mg cya-3-gluE/g (*L**: 75.27, *a**: 11.20, *b**: 9.55) After 4 weeks of storage 0.039 mg cya-3-gluE/g (*L**: 75.19, *a**: 11.35, *b**: 9.75)	[53]
		Yogurt	Eggplant	Color parameters of the yogurt with 1% of encapsulated extract: *L**: 83.74, *a**: 3.96 *b**: 8.18.	[159]
		Bakery products	Onion skin	Color parameters of the extracts: *L**: 22.56, *a**: 20.84, *b**: −0.78.	[160]
		Marmalade	Black carrot	Thermal processing caused a loss of anthocyanins (79.2–89.5%)	[161]
		Red beer	Eggplant peel extract	Color parameters: *L**: 74.67, *a**: 9.86, *b**: 21.76	[162]
	Carotenoids (lycopene)	Butter	Tomato skin	Extract concentration: 20 mg/kg Color parameters, at initial time: *L**: 48.78, *a**: 2.36, *b**: 14.17 Color parameters at 4 months of storage at 4–6 °C: *L**: 48.37, *a**: 2.25, *b**: 14.18	[163]
		Mayonnaise	Tomato skin	Extract concentration: 50 mg/kg Color parameters, at initial time: *L**: 55.82, *a**: 10.32, *b**: 19.95 Color parameters at 4 months of storage at 4–6 °C: *L**: 53.66, *a**: 9.64, *b**: 19.10	[163]
		Ice cream	Tomato skin	Extract concentration: 70 mg/kg Color parameters, at initial time: *L**: 84.41, *a**: 11.56, *b**: 28.59 Color parameters at 4 months of storage at −25 °C: 83.97, *a**: 11.16, *b**: 21.29	[163]
		Ice cream	Tomato peel	The ice cream prepared with 3% of extract obtained the highest score in all the parameters of the sensorial analysis	[91]
Pink	Betalains	Gummy candies	Cactus (*Opuntia ficus-indica* L. Mill.) fruit pulp	High stability of the color in the gummy candies were observed after 30 days of storage al 4 °C	[164]
		Yogurt	Pitaya	The yogurts with 0.5% and 2% of the colorant were statistically similar to the commercial yogurts with beetroot and carmine colorants, respectively	[98]
		Cookies	*Gomphrena globose* L.	Color parameters of samples with the lyophilized extract: *L**: 56.3, *a**: 22.1, *b**: 8.0 Color parameters of samples with the spray dried extract: *L**: 56.0, *a**: 25.5, *b**: 4.3	[165]
		Ice cream	*Gomphrena globose* L.	Color parameters: *L**: 86, *a**: 8, *b**: 2.4	[166]
		Tagliatelle pasta	Flowers of Amaranthus caudatus	Color parameters: *L**: 62, *a**: 17, *b**: 8	[166]
		Meringue cookies	Red-fleshed pitaya peels	Color parameters: *L**: 79.6, *a**: 14.5, *b**: 1.24	[166]
Yellow-Orange	Carotenoids	Dough biscuits	Mango peel	Color parameters of product with 20% of mango peel: *L**: 52.90, *a**: 7.71, *b**: 22.02	[167]
		Macaroni	Mango peel	-	[167]
Green	Chlorophylls	Syrup	*Centella asiatica* L.	The syrups with the extracts treated with zinc and copper were stable and presented minimum change in color, mostly an increase in the *h** value.	[148]
		Bread	*Centella asiatica* L.	The bread with the extract treated with zinc presented a yellow–green color, while the one with the extract treated with copper presented a green color. All of them changed significantly after 7 days	[148]
		White chocolate	*Nannochloropsis oculata* D.J. Hibberd	Samples with the encapsulated extract presented a higher quantity of chlorophylls. However, the samples with free extract presented a better stability of the color along the 28 days	[168]
		Chewing gum	*Isochrysis galbana* Pascher	Samples showed a decrease in hardness as well as in the cohesiveness, which is advantageous for a chewing gum	[169]
		Yogurt	Spirulina	The preparation with 0.25% showed the best acceptability in the sensory analysis	[170]
Blue	Anthocyanins	Cooked rice	Butterfly pea flowers	The incorporation of 0.6% of extract resulted in better acceptance by consumers	[154]

*L**: lightness; *a**: red/green coordinate; *b**: yellow/blue coordinate; *h*: hue; *C**: chroma.

## Data Availability

Not applicable.

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
