# Peer review of "Natural Sources of Food Colorants as Potential Substitutes for Artificial Additives"

_foods, 2023, doi:10.3390/foods12224102_

Round 1
Reviewer 1 Report
Comments and Suggestions for Authors
The subject of the manuscript is extremely interesting in the context of increasing interest in increasing the nutritional value and safety of food. However, some additions are necessary:
- a presentation of the database analyzed for the creation of the manuscript
- a presentation of the limits determined by the possible interactions or adverse effects for each type of dye presented when incorporated in dierite food products, limits of use, restrictions determined by interactions with different types of food matrices
Author Response
The subject of the manuscript is extremely interesting in the context of increasing interest in increasing the nutritional value and safety of food.
Answer: The authors thank the reviewer for the valuable comments leading to improve the manuscript. A carefully revision of the paper was performed considering the reviewer comments. All changes were highlighted all over the manuscript.
However, some additions are necessary:
- a presentation of the database analyzed for the creation of the manuscript
Answer: Accordingly, to the Reviewer comment, in the abstract is mentioned some parameters of the data collection and at the end of the introduction was written the way it was designed (how data was collected, among others) (lines 106 – 117).
- a presentation of the limits determined by the possible interactions or adverse effects for each type of dye presented when incorporated in dierite food products, limits of use, restrictions determined by interactions with different types of food matrices
Answer: The authors thank the reviewer comment. In the European Union, the limit of use, restrictions or food matrix formulation conditions of food additives are regulated by the European regulation (EU) nº 1129/2011. In which, the food colourants are classified as “Food colours authorised at quantum satis” (whose content is not limited, quantum satis mean that no maximum numerical level is specified and substances shall be used in accordance with good manufacturing practice, at a level not higher than is necessary to achieve the intended purpose and provided the consumer is not misled) or “Food colours with combined maximum limit” (whose use is restricted in term of dose and food matrix). This appreciation was included along the manuscript when any authorized food additive was mention.
Reviewer 2 Report
Comments and Suggestions for Authors
This research is important and brings valuable information for further research and with practical application. The presented research is well-planned, and the manuscript is generally well organized.
Therefore, the work could be of interest, but some points must be considered:
The class of anthraquinones with interesting representations could also be listed in the Abstract.
In the Introduction part, more details related to the structural classes of interest would be needed, with the general structures of the structural classes of interest and some examples.
line 100-101- The presentation of the general structure of anthocyanins is not accurate - it must be corrected
More information related to color parameters would be necessary - the parameters are presented only as symbols.
In the tables, some names of the species should be corrected and the abbreviations should be explained in the Notes.
The authors could highlight deficient aspects in this research field which could be addressed in their future research. In Conclusions is necessary to present the limitations and deficiencies of current studies and perspectives for future studies that could be implemented to cover them.
Author Response
This research is important and brings valuable information for further research and with practical application. The presented research is well-planned, and the manuscript is generally well organized.
Answer: The authors thank the reviewer for the valuable comments leading to improve the manuscript. A carefully revision of the paper was performed considering the reviewer comments. All changes were highlighted all over the manuscript.
Therefore, the work could be of interest, but some points must be considered:
The class of anthraquinones with interesting representations could also be listed in the Abstract.
Answer: following the reviewer suggestions, anthraquinones were also listed in the abstract as part of representative compounds that present colorant capacity.
In the Introduction part, more details related to the structural classes of interest would be needed, with the general structures of the structural classes of interest and some examples.
Answer: the authors thank the reviewer comment. Following the suggestion, the introduction was improved accordingly.
line 100-101- The presentation of the general structure of anthocyanins is not accurate - it must be corrected.
Answer: The anthocyanins structure description was carefully revised and rewritten accordingly (following the previously reported by Kong et al., 2003; Morata et al., 2019 and Wolfe et al., 2008).
More information related to color parameters would be necessary - the parameters are presented only as symbols.
Answer: L*, a* and b* are the CIELAB parameters. A brief explanation regarding CIELAB method (and its parameters) was included in the manuscript (Line 285 – 287)
In the tables, some names of the species should be corrected and the abbreviations should be explained in the Notes.
Answer: Following the reviewer recommendation the names of the species were corrected and the abbreviation in the tables were explained with a food note
The authors could highlight deficient aspects in this research field which could be addressed in their future research. In Conclusions is necessary to present the limitations and deficiencies of current studies and perspectives for future studies that could be implemented to cover them.
Answer: Following the reviewer suggestion the conclusion was improved mentioning and highlighting several deficiencies in the area, the future perspectives of the field, as well as suggesting improvement actions that could be followed.
Reviewer 3 Report
Comments and Suggestions for Authors
The work submitted for review addresses an important topic for the food industry, namely dyes added to food. The authors of the publication have collected information on this subject, which, however, requires supplementation and improvement. Below I have presented my comments and suggestions on how to improve this manuscript.
1. The title of the work raises objections: New sources of natural food colourants as potential replacements of their artificial counterparts. A review. -It is not very clear what new sources of natural dyes we are talking about? In my opinion, the word new should be removed. Maybe it would be better to change the title to: Natural sources of food dyes as potential substitutes for artificial additives.
2. Abstract:
-Very long sentence at the end of Abstract should be rewritten :”Therefore, the present review compiles the novel sources of these colourant compounds, referring to their obtention, identification and some of the efforts made for the improvement of their stability, likewise, it collects the result of the introduction of the colourants obtained in different food matrices, evidencing in this way, the promising path of development of natural colourants for the replacement of their artificial counterparts.”
-Moreover, the expression "the novel sources" is unjustified. The use of the word "various sources" is true.
-Since this is a review article, the abstract should at least include what type of review it is, what years were taken into account in collecting the literature, and what databases were searched.
3. Introduction
Line 22. Please explain the abbreviation "FDA".Typically, you should explain abbreviations when they first appear.
Line 44. “Another way is as straight colours, lakes, and mixtures.”- lakes- it seems like mistake.
Line 50 “Currently, artificial food colourants are widely use by the food industry, especially in children’s products, since they have high intensity, stability, uniformity of colour, and are cheap.”-it looks like artificial food colourants have been used especially, intentionally as additives for children’s products. Please change “especially” into among others or also.
Line 55” Since then, several studies have been published, the most known being the one made by McCann [6] who evidenced the increase in the incidence of ADHD including inattention, impulsivity, and overactivity not only in children with extreme hyperactivity, but also in the general child population due to the consumption of artificial food colour (AFCs) and other additives”- Please list the other food additives that caused neurological disorders in children so that it does not look as if artificial colors were the main toxin.
Line 60-“ Consequently, the responsible entities carried out different reviews of the available 60 studies in order to evaluate the safety of azo colourants, among other additives”- please write which entities and in which countries.
Line 62- "On one hand, the most important communications have been the evaluation carried out in 2009 where only a decrease of the acceptable daily intakes" - who issued this message, in which country? reference from 2009 is needed.
Line 66 “and the statement release in 2013 where it was assured that a revaluation of the ADI of any of the azo colourants was not necessary and it was recommended carry out new tests related to genotoxicity[3].”-Ref 3 it is “EFSA European Food Safety Authority Available online: https://www.efsa.europa.eu/en/topics/topic/food-colours (accessed 1012 on 3 May 2022).”- you must correctly describe the citation and mention in the text who issued the statement in question.
In the introduction, the authors focused on the harmfulness of artificial dyes to children, while ignoring the rest of the population. Isn't there such research? If it has not been done for adults, please mention it.
At the end of the introduction, you should indicate what this review is about and how the data was collected.
4. Chapter 2
Figure 1. should be at the end of chapter 2 before 2.1. Information about the program used to draw the patterns can be omitted.
The subsections for Chapter 2 are very general. The information contained there is too general for a scientific work. The work concerns natural dyes and in my opinion this part of the work should be developed.
5. Chapter 3.
Line 167 Explain the abbreviation EFSA.
Table 1 Explain: CFR , Obtention-change into isolation
Line 222. Please explain the meaning of L*: a*, b*
Line 245:” Moreover, the anthocyanin extract obtained (21.99 mg of monomeric anthocyanin/ L) was stabilized by three different ways where the use of buffer combination (sodium carbonate and sodium bicarbonate)”-please add pH value of applied buffer system.
Table 2-Please add more details concerning Extraction conditions. For example” acidified methanol.”-what is the concentration of acid? What kind of acid do you mean? and so on.
Table 3 and Table 4 .Three columns titled: Extraction conditions/Optimum extraction conditions – should be combined.” colour parameters”- should be a separate column.
Table 4 Column “Carotenoids”-should be deleted, and the table’totle should be appropriate modified.
6. In the conclusion, the authors should indicate what possibilities they see for further research. What should be the direction of research? Where do they see potential for development?
Author Response
The work submitted for review addresses an important topic for the food industry, namely dyes added to food. The authors of the publication have collected information on this subject, which, however, requires supplementation and improvement. Below I have presented my comments and suggestions on how to improve this manuscript.
Answer: The authors thank the reviewer for the valuable comments leading to improve the manuscript. A carefully revision of the paper was performed considering the reviewer comments. All changes were highlighted all over the manuscript.
- The title of the work raises objections: New sources of natural food colourants as potential replacements of their artificial counterparts. A review. -It is not very clear what new sources of natural dyes we are talking about? In my opinion, the word new should be removed. Maybe it would be better to change the title to: Natural sources of food dyes as potential substitutes for artificial additives.
Answer: Following the reviewer recommendation, the title was appropriately modified to “Natural sources of food colourants as potential substitutes for artificial additives”
- Abstract:
- Very long sentence at the end of Abstract should be rewritten :”Therefore, the present review compiles the novel sources of these colourant compounds, referring to their obtention, identification and some of the efforts made for the improvement of their stability, likewise, it collects the result of the introduction of the colourants obtained in different food matrices, evidencing in this way, the promising path of development of natural colourants for the replacement of their artificial counterparts.”
Answer: The sentences was revised and rewritten (lines 23 – 27).
- Moreover, the expression "the novel sources" is unjustified. The use of the word "various sources" is true.
Answer: Following the reviewer comment, the sentence “the novel sources” was appropriately modified to “various food sources”.
- Since this is a review article, the abstract should at least include what type of review it is, what years were taken into account in collecting the literature, and what databases were searched.
Answer: The authors thank the reviewer appreciations. This useful information was included in the manuscript Lines 22 – 23.
- Introduction
Line 22. Please explain the abbreviation "FDA". Typically, you should explain abbreviations when they first appear.
Answer: FDA means Food and Drug Administration; this information was included in the text the first time the abbreviation was cited (line 34)
Line 44. “Another way is as straight colours, lakes, and mixtures.”- lakes- it seems like mistake.
Answer: It´s not a mistake, the FDA definition of Colour additives said: “Color additives are classified as straight colors, lakes, and mixtures. Straight colors are color additives that have not been mixed or chemically reacted with any other substance (for example, FD&C Blue No. 1 or Blue 1). Lakes are formed by chemically reacting straight colors with precipitants and substrata (for example, Blue 1 Lake). Lakes for food use must be made from certified batches of straight colors. Lakes for food use are made with aluminum cation as the precipitant and aluminum hydroxide as the substratum”. The sentence was revised and rewritten with a couple of examples for a better understanding.
Line 50 “Currently, artificial food colourants are widely use by the food industry, especially in children’s products, since they have high intensity, stability, uniformity of colour, and are cheap.”-it looks like artificial food colourants have been used especially, intentionally as additives for children’s products. Please change “especially” into among others or also.
Answer: Following the suggestion, the sentences was rewritten (lines 52 – 53).
Line 55” Since then, several studies have been published, the most known being the one made by McCann [6] who evidenced the increase in the incidence of ADHD including inattention, impulsivity, and overactivity not only in children with extreme hyperactivity, but also in the general child population due to the consumption of artificial food colour (AFCs) and other additives”- Please list the other food additives that caused neurological disorders in children so that it does not look as if artificial colors were the main toxin.
Answer: The sodium benzoate (E 211) was also included in the manuscript as a potential food additive involve in the ADHD study performed by McCann et al. 2007 (Line 61 – 62).
Line 60-“Consequently, the responsible entities carried out different reviews of the available 60 studies in order to evaluate the safety of azo colourants, among other additives”- please write which entities and in which countries.
Answer: The responsible entities that carry out the mentioned reviews was EFSA and FDA. This information was appropriately included in the manuscript following the reviewer suggestion (Lines 77 and 78).
Line 62- "On one hand, the most important communications have been the evaluation carried out in 2009 where only a decrease of the acceptable daily intakes" - who issued this message, in which country? reference from 2009 is needed.
Answer: Following the reviewer comment, this sentence was rewritten including the required information (line 81) and the reference was added (line 87)
Line 66 “and the statement release in 2013 where it was assured that a revaluation of the ADI of any of the azo colourants was not necessary and it was recommended carry out new tests related to genotoxicity[3].”-Ref 3 it is “EFSA European Food Safety Authority Available online: https://www.efsa.europa.eu/en/topics/topic/food-colours (accessed 1012 on 3 May 2022).”- you must correctly describe the citation and mention in the text who issued the statement in question.
Answer: The reviewer comment was followed (line 85, 87) and the reference was corrected.
In the introduction, the authors focused on the harmfulness of artificial dyes to children, while ignoring the rest of the population. Isn't there such research? If it has not been done for adults, please mention it.
Answer: To the authors knowledge, very few studies regarding the effect of artificial food colourants in adults have been made. The studies performed by Murdoch et al., Di Lorenzo et al., Pestana et al. and Park et al., were described and included in the manuscript (line 62 – 72)
At the end of the introduction, you should indicate what this review is about and how the data was collected.
Answer: Accordingly, to the Reviewer comment, the end of the introduction was rewritten clarifying the main purpose of this review and the way it was designed (how data was collected, among others) (lines 106 – 117).
- Chapter 2
Figure 1. should be at the end of chapter 2 before 2.1. Information about the program used to draw the patterns can be omitted.
Answer: This information was omitted and figure 1 was moved following the reviewer suggestion.
The subsections for Chapter 2 are very general. The information contained there is too general for a scientific work. The work concerns natural dyes and in my opinion this part of the work should be developed.
Answer: The authors are grateful for the reviewer suggestion, although from our point of view, the information given in this section (structure, stability, among others) introduce the relevance of topic. Subsections of chapter 2 was carefully revised and improved including relevant information about the mentioned colourants.
- Chapter 3.
Line 167 Explain the abbreviation EFSA.
Answer: EFSA means European Food Safety Agency, this information was included in the text the first time it was mention. It was also included in line 77.
Table 1 Explain: CFR, Obtention-change into isolation
Answer: CFR means “Code of Federal Regulations”, this information was included as a foot note in the table. Moreover, “Obtention” was changed to “isolation” following the reviewer comments.
Line 222. Please explain the meaning of L*: a*, b*
Answer: L*, a* and b* are the CIELAB parameters. A brief explanation regarding CIELAB method (and its parameters) was included in the manuscript (Line 285 – 287)
Line 245:” Moreover, the anthocyanin extract obtained (21.99 mg of monomeric anthocyanin/ L) was stabilized by three different ways where the use of buffer combination (sodium carbonate and sodium bicarbonate)”-please add pH value of applied buffer system.
Answer: Following the reviewer suggestion, the information of the pH values of applied buffer system was included in the manuscript for a better understanding (Line 312).
Table 2-Please add more details concerning Extraction conditions. For example” acidified methanol.”-what is the concentration of acid? What kind of acid do you mean? and so on.
Answer: Table 2 was improved according to the reviewer suggestion. The information concerning the extraction conditions was included when such information was available (not always).
Table 3 and Table 4. Three columns titled: Extraction conditions/Optimum extraction conditions – should be combined.” colour parameters”- should be a separate column.
Answer: The reviewer suggestions regarding tables 3 and 4 was follow.
Table 4 Column “Carotenoids”-should be deleted, and the table’s title should be appropriate modified.
Answer: The authors thank the reviewer suggestion. The table´s title was slightly modified (including the term “carotenoids and betaxanthins”). However, we consider not to delate the column “pigment” in order to continue with the other tables stile. Moreover, we include novel information in this table related to betaxanthins.
- In the conclusion, the authors should indicate what possibilities they see for further research. What should be the direction of research? Where do they see potential for development?
Answer: Following the reviewer suggestion the conclusion was improved mentioning the future perspectives of the field, as well as suggesting improvement actions that could be followed
Round 2
Reviewer 1 Report
Comments and Suggestions for Authors
Accept the present form
Reviewer 3 Report
Comments and Suggestions for Authors
The manuscript has been revised in accordance with my suggestions and therefore I recommend it for publication.